J Physiol 603.20 (2025) pp 6391–6421

6391

# Kif1a and intact microtubules maintain synaptic-vesicle populations at ribbon synapses in zebrafish hair cells

Sandeep David[1,2] (ID), Katherine Pinter[1], Keziah-Khue Nguyen[3,4], David S. Lee[3] (ID), Zhengchang Lei[1], Yuliya Sokolova[5] (ID), Lavinia Sheets[3,4] (ID) and Katie S. Kindt[1] (ID)

[1] Section on Sensory Cell Development and Function, National Institute on Deafness and other Communication Disorders, Bethesda, Maryland, USA
[2] National Institutes of Health–Brown University Graduate Partnership Program, Bethesda, Maryland, USA
[3] Department of Otolaryngology, Head and Neck Surgery, Washington University School of Medicine, St. Louis, Missouri, USA
[4] Department of Developmental Biology, Washington University School of Medicine, St. Louis, Missouri, USA
[5] Advanced Imaging Core, National Institute on Deafness and other Communication Disorders, Bethesda, Maryland, USA

Handling Editors: Katalin Toth & Samuel Young

The peer review history is available in the Supporting Information section of this article (https://doi.org/10.1113/JP286263#support-information-section).

**Abstract figure legend** Sensory hair cells of the inner ear and lateral-line system exhibit strong synaptic-vesicle enrichment at the presynapse, around specialized structures called ribbons. We find that hair cells in the zebrafish lateral-line system rely on the kinesin motor protein Kif1a and microtubules to enrich synaptic vesicles at presynaptic ribbons. Light and electron microscopy reveal that *kif1aa* mutants have fewer synapses and fail to properly enrich synaptic vesicles at the ribbons. Functional assessments show that *kif1aa* mutants have reduced spontaneous spikes at the afferent cell body and smaller evoked calcium responses at the afferent terminals compared with controls. These synaptic deficiencies significantly impact the zebrafish's ability to rheotax, a behaviour that requires the lateral-line system to orient against water flow. While wild-type zebrafish maintain their position in flow, *kif1aa* mutants are unable to sustain this behaviour.

This article was first published as a preprint. David S, Pinter K, Nguyen K-K, Lee DS, Lei Z, Sokolova Y, Sheets L, Kindt KS. 2024. Kif1a and intact microtubules maintain synaptic-vesicle populations at ribbon synapses in zebrafish hair cells. bioRxiv. https://doi.org/10.1101/2024.05.20.595037

The Journal of Physiology

**Abstract** Sensory hair cells of the inner ear utilize specialized ribbon synapses to transmit sensory stimuli to the central nervous system. This transmission necessitates rapid and sustained neurotransmitter release, which depends on a large pool of synaptic vesicles at the hair-cell pre-synapse. While previous work in neurons has shown that kinesin motor proteins traffic synaptic material along microtubules to the presynapse, the mechanisms of this process in hair cells remain unclear. Our study demonstrates that the kinesin motor protein Kif1a, along with an intact microtubule network, is essential for enriching synaptic vesicles at the presynapse in hair cells. Through genetic and pharmacological approaches, we disrupt Kif1a function and impair micro-tubule networks in hair cells of the zebrafish lateral-line system. These manipulations led to a significant reduction in synaptic-vesicle populations at the presynapse in hair cells. Using electron microscopy, *in vivo* calcium imaging, and electrophysiology, we show that a diminished supply of synaptic vesicles adversely affects ribbon-synapse function. *Kif1aa* mutants exhibit dramatic reductions in spontaneous vesicle release and evoked postsynaptic calcium responses. Furthermore, *kif1aa* mutants exhibit impaired rheotaxis, a behaviour reliant on the ability of hair cells in the lateral line to respond to sustained flow stimuli. Overall, our results demonstrate that Kif1a-mediated microtubule transport is critical to enrich synaptic vesicles at the active zone, a process that is vital for proper ribbon-synapse function in hair cells.

(Received 20 May 2024; accepted after revision 5 September 2024; first published online 4 October 2024)

**Corresponding author** K. S. Kindt: Section on Sensory Cell Development and Function, National Institute on Deafness and other Communication Disorders, Bethesda, MD, USA. Email: katie.kindt@nih.gov

**Key points**

- *Kif1a* mRNAs are present in zebrafish hair cells.
- Loss of Kif1a disrupts the enrichment of synaptic vesicles at ribbon synapses.
- Disruption of microtubules depletes synaptic vesicles at ribbon synapses.
- *Kif1aa* mutants have impaired ribbon-synapse and sensory-system function.

## Introduction

Sensory hair cells in the inner ear convert sensory stimuli into electrical impulses, which are transmitted to the brain. When sensory stimuli deflect hair bundles at the cell's apex, mechanoelectrical transduction channels open, depolarizing the cell in a graded manner (Gillespie & Walker, 2001). Cell depolarization activates voltage-gated $Ca_v1.3$ channels at the presynaptic active zone (AZ) at the base of the hair cell (Brandt et al., 2003). The resulting calcium influx through these channels triggers synaptic-vesicle fusion and the release of glutamate onto afferent neurons. In hair cells, neurotransmission is both rapid and sustained, due to a specialized structure known as the synaptic ribbon (Matthews & Fuchs, 2010; Moser et al., 2006), which anchors synaptic vesicles at the presynaptic AZ, readying them for release. To supply these vesicles, hair cells have efficient mechanisms to recycle synaptic vesicles at the AZ in order to sustain release (Pangrsic & Vogl, 2018; Wichmann & Moser, 2015). Importantly, disrupting presynapse morphology or vesicle recycling disrupts ribbon-synapse function, which can lead to hearing and balance impairments (Jing et al., 2013; Kroll et al., 2019; Trapani et al., 2009; Wan et al.,

**Sandeep David** is a PhD student in Neuroscience in the National Institutes of Health – Brown University Graduate Partnership Program under the supervision of Dr Katie Kindt in the National Institute on Deafness and Communicative Disorders. He received his Bachelor of Science in Physiology and Neurobiology at the University of Maryland, College Park. His research focuses on sensory hair cells, synaptic-vesicle populations and synapse function, using zebrafish as an animal model.

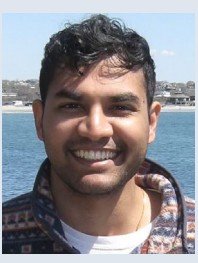

2019). While local vesicle recycling is important for ribbon-synapse function, much less is known about how new synaptic material is synthesized and transported to the presynaptic AZ in hair cells.

Synaptic-vesicle recycling, through endocytosis and repackaging of synaptic materials at the presynaptic AZ, has been shown to be critical for maintaining synapse function in neurons and hair cells (Gallimore et al., 2023; Neef et al., 2014). But in neurons, numerous studies have also outlined the importance of *de novo* transport of synaptic material from the cell soma to presynaptic terminals (Rizzoli, 2014; Santos et al., 2009). In the soma of neurons, new synaptic material is synthesized in the endoplasmic reticulum (ER) and then packaged and sorted in the Golgi. This material leaves the Golgi as synaptic cargos that are transported along axons by kinesin motor proteins via microtubule highways. Kinesins transport cargo, such as synaptic-vesicle precursors, towards the plus-end of microtubules. In neurons, the plus-end of microtubules in axonal nerve processes are oriented towards the presynapse; thus, kinesin-mediated transport delivers synaptic cargos to the presynapse (Guedes-Dias & Holzbaur, 2019). KIF1A, a member of the kinesin-3 family, has been shown to play a conserved role in transporting synaptic-vesicle precursors to neuronal presynapses across multiple species (Pack-Chung et al., 2007; Rizalar et al., 2021; Yonekawa et al., 1998; Zhao et al., 2001). In neurons, axons can be extremely long, and cargo must be transported considerable distances to reach the presynapse. In contrast, hair cells are compact, polarized epithelial cells. In hair cells, the Golgi is located above the nucleus, and the presynaptic AZ is just below the nucleus (Fig. 1*D*; Siegel & Brownell, 1986). Recent work has demonstrated that hair-cell microtubules grow their plus ends from the cell apex, past the Golgi, and toward the presynaptic AZ (Hussain et al., 2024; Voorn et al., 2024). Additionally, immunostaining and scRNAseq data have shown that Kif1a is present in hair cells (Michanski et al., 2019; Sur et al., 2023). However, whether Kif1a or a microtubule network is required for proper transport of synaptic-vesicle precursors to the presynaptic AZ in hair cells remains unknown.

To study the role of Kif1a and microtubules in synaptic-vesicle transport in hair cells, we examined hair cells in larval zebrafish. In zebrafish, hair cells are found in the inner ear and in the lateral-line system, which are responsible for hearing and balance, and detection of local water flow, respectively (Fig. 1*A*; Nicolson, 2005; Sheets et al., 2021). In larval zebrafish, the inner ear is composed of five sensory organs (three cristae and two maculae), while the lateral line is made up of neuromast organs that form in lines along the fish. Both of these sensory systems develop rapidly in zebrafish and are mature when larvae are only 5 days post-fertilization (dpf) (Bhandiwad et al., 2013; Zeddies & Fay, 2005). Importantly, hair cells in the larval zebrafish are morphologically, genetically, and functionally similar to mammalian hair cells, and many of the core synaptic molecules are conserved between zebrafish and mammals. For example, in both mammals and zebrafish: ribeye makes up the majority of the ribbon density, Vglut3 is the vesicular glutamate transporter required to uptake glutamate into synaptic vesicles, and calcium influx through $Ca_v1.3$ channels triggers vesicle fusion (Fig. 1*E*; Lv et al., 2016; Obholzer et al., 2008; Sidi et al., 2004). The lateral-line system has proved particularly advantageous for the study of hair-cell synapses due to the superficial location of neuromasts along the body and the transparency of zebrafish at larval stages. Previous research using both fixed and live imaging approaches in lateral-line organs of larval zebrafish has shown that synaptic vesicles are highly enriched at the base of lateral-line hair cells (Fig. 1*C*; Einhorn et al., 2012; Obholzer et al., 2008). In addition, there are powerful transgenic lines to visualize ribbon synapses or monitor ribbon-synapse activity *in vivo* using genetically encoded indicators (Zhang et al., 2018). To complement this functional imaging, established electrophysiological approaches make it possible to measure vesicle fusion by recording spikes from the afferent neurons that innervate lateral-line neuromasts (Trapani & Nicolson, 2011). Together, these experimental advantages and tools make zebrafish an excellent model for studying the role of synaptic-vesicle transport at ribbon synapses in hair cells.

In our study, we identify Kif1a as a kinesin motor protein important to transport and ultimately enrich synaptic vesicles at the presynaptic AZ in hair cells. We show that while both zebrafish paralogues of mammalian *kif1a*, *kif1aa* and *kif1ab* are expressed in inner-ear hair cells, only *kif1aa* is expressed in lateral-line hair cells. We find that *kif1aa* mutants are unable to enrich synaptic vesicles at the presynaptic AZ of lateral-line hair cells. Likewise, wild-type zebrafish exhibit the same loss of synaptic-vesicle enrichment when microtubules are destabilized. Functionally, we show that *kif1aa* mutants have normal evoked presynaptic calcium responses but that postsynaptic calcium responses in afferent neurons are dramatically reduced. Further, the spontaneous spike rate of lateral-line afferent neurons is decreased in *kif1aa* mutants. Behaviourally, we find that *kif1aa* mutants have impaired station holding behaviour during rheotaxis – a behaviour that depends on a functional lateral-line system. Overall, our work demonstrates that a Kif1a-based mechanism enriches synaptic vesicles at ribbon synapses in hair cells. This enrichment is essential for ribbon-synapse function and ultimately for sensory behaviour.

## Methods

### Zebrafish husbandry

Zebrafish (*Danio rerio*) lines were maintained at the National Institutes of Health (NIH) under animal study protocol #1362-13. Zebrafish larvae (0–6 dpf) were raised in incubators at 28°C in E3 embryo medium (E3: 5 mM NaCl, 0.17 mM KCl, 0.33 mM $CaCl_2$ and 0.33 mM $MgSO_4$, buffered in HEPES, pH 7.2) with a 14 h:10 h light:dark cycle. Lines were maintained in a Tu or TL background. All experiments were conducted on larvae at 3–6 dpf. During this period, larvae received nutrition from their

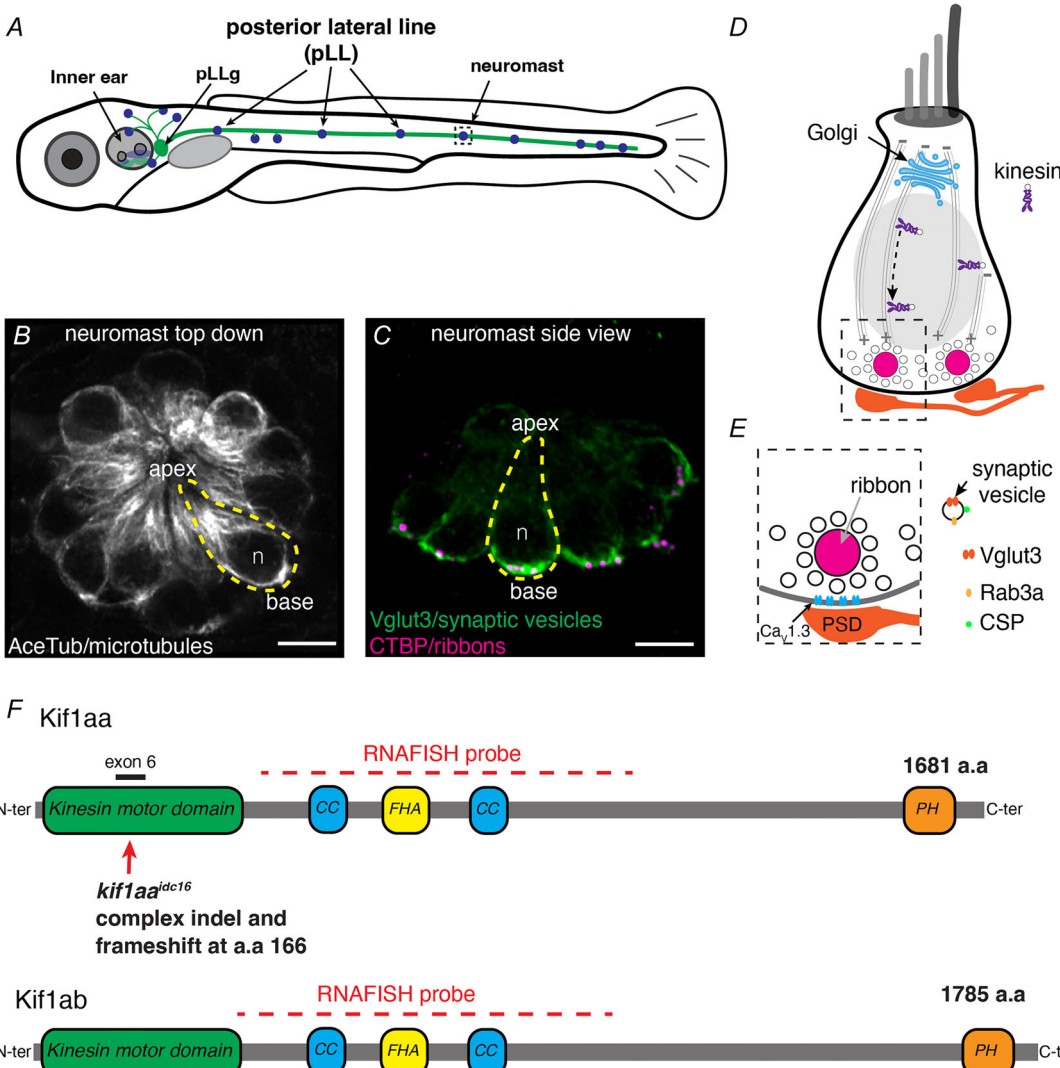

**Figure 1. The zebrafish lateral-line system and *Kif1a* paralogues**

*A*, schematic of a larval zebrafish at 5 dpf. Hair cells are present in the inner ear and lateral line (blue). Hair cells are innervated by neurons in the posterior lateral-line ganglion (pLLg) located near the inner ear (green). *B–C*, the lateral line is composed of clusters of hair cells called neuromasts (5 dpf). The apices of hair cells project from the centre of these clusters, while the ribbon synapses are located at the base of the cells. Neuromasts can be viewed from the top down (*B*) or from the side (*C*). An individual hair cell in *B* and *C* is outlined in yellow. Within hair cells, microtubule networks extend along the apical–basal axis (*B*). At the base of hair cells, synaptic vesicles (Vglut3 label, green) are enriched near the presynapse or ribbons (CTBP, magenta) (*C*). *D*, within hair cells, the Golgi is located above the nucleus (grey oval, n in *B–C*). The Golgi is where synaptic-vesicle precursors are made *de novo*. Kinesin motors could be used to transport vesicles along microtubules to the cell base. *E*, synaptic vesicles surround the presynapse or ribbon in hair cells. Synaptic vesicles contain Rab3a, CSP and Vglut3. Beneath the ribbon $Ca_V 1.3$ channels are clustered adjacent to the postsynaptic density (PSD). *F*, overview of the Kif1aa and Kif1ab proteins and major domains (coiled coil (CC), fork-head associated (FHA), pleckstrin homology (PH)). The location of the germline *kif1aa* lesion in the kinesin motor domain within exon 6 is indicated. The red dashed line indicates the regions encompassed by the RNA-FISH probe. The scale bar in *B* and *C* = 5 µm.

yolk and were not fed. One exception to this feeding regimen was larvae assayed for rheotaxis. These larvae were provided with additional food to ensure consistent behavioural outcomes in our rheotaxis assays. No animals died before the conclusion of our experiments. To kill them, larvae were rapidly cooled on ice and prepared for immunohistochemistry or cooled on ice and then placed in a $-20°C$ freezer at the end of the experiment.

For rheotaxis experiments, embryos were shipped at 1 dpf from NIH to Washington University–St. Louis and maintained under Institutional Animal Care and Use Committee protocol #23-0078, and subsequently raised in incubators at 28°C in embryo media2 (EM2: 15 mM NaCl, 0.5 mM KCl, 1 mM CaCl$_2$, 1 mM MgSO$_4$, 0.15 mM KH$_2$PO$_4$, 0.042 mM Na$_2$HPO$_4$, 0.714 mM NaHCO$_3$) with a 14 h:10 h light:dark cycle. After 4 dpf, larvae were raised in 100–200 ml of EM2 in 250 ml plastic beakers and fed rotifers daily.

Neuromasts L1–L4 were used for all analyses, except the Ca$_V$1.3 immunolabel where L1–L4, D1 and O1 were examined, and our TEM analysis where cranial neuromasts M1, along with the POs, and SOs were examined. Larvae were chosen at random and used at an age where sex determination is not possible. The previously described mutant and transgenic lines were used in this study: *kif1aa*$^{idc24}$, *Tg(myo6b:memGCaMP6s)*$^{idc1Tg}$, *Tg(en.sill,hsp70l:GCaMP6s)*$^{idc8Tg}$, *Tg(myo6b:Cr.ChR2-EYFP)*$^{ahc1Tg}$, *Tg(myo6b:ctbp2-TagRFP)*$^{idc11Tg}$ (also referred to as: *Tg(myo6b:rib a-TagRFP)*$^{idc11Tg}$) and *Tg(myo6b:YFP-Hsa.TUBA)*$^{idc16Tg}$ (Hussain et al., 2024; Monesson-Olson et al., 2014; Ohta et al., 2020; Wong et al., 2019; Zhang et al., 2018). These lines are also described on ZFIN: ZDB-ALT-240416-3, ZDB-ALT-170113-3, ZDB-ALT-171206-1, ZDB-ALT-140924-1, ZDB-ALT-190102-4 and ZDB-ALT-210824-2).

The *kif1aa* germline mutant (ZDB-ALT-240416-3) has been previously described (Hussain et al., 2024). This Crispr-Cas9 mutant has a complex insertion-deletion that leads to a disruption in Kif1aa at amino acid 166, in exon 6, within the kinesin motor domain (Fig. 1*F*). This insertion-deletion also disrupts a Bsl1 restriction site in exon 6. Genotyping was done using standard PCR followed by a Bsl1 restriction enzyme digest. *Kif1aa* genotyping primers used were as follows: *kif1aa*_FWD 5'-AACACCAAGCTGACCAGTGC-3' and *kif1aa*_REV 5'-TGCGGTCCTAGGCTTACAAT-3'. *Kif1aa* mutants were compared with either heterozygous and wild-type siblings (sibling controls) or to wild-type siblings as stated in the text.

## Nocodazole treatment

To destabilize microtubules, *Tg(myo6b:YFP-Hsa.TUBA)*$^{idc16Tg}$ larvae at 5 dpf were incubated in 250 nM nocodazole (Sigma-Aldrich, SML1665) for 2 h. This concentration

of nocodazole has previously been shown to disrupt microtubules in zebrafish hair cells (Hussain et al., 2024). Nocodazole was diluted in E3 media for a final concentration of 250 nM nocodazole and 0.1% DMSO. After effective nocodazole treatment, *YFP-Hsa.TUBA* labelling was visually disrupted. For controls, larvae were incubated in media containing 0.1% DMSO. After 2 h, larvae were fixed for immunohistochemistry or prepared for LysoTracker labelling (see below).

## Lysotracker labelling and imaging

After 2 h of nocodazole treatment, 100 nM LysoTracker Red DND-99 (ThermoFisher, L7528) was added to the media for 15 min. This concentration and duration has previously been used to label synaptic vesicles in zebrafish hair cells (Einhorn et al., 2012). Larvae were then embedded in 1% low melt agarose prepared in E3 media containing 0.03% Tricaine-S (SYNCAINE/MS-222, Syndel), 100 nM LysoTracker, and either 250 nM nocodazole or 0.1% DMSO. A similar labelling approach was used for LysoTracker labelling in *kif1aa* mutants. LysoTracker Red DND-99 was used to label *Tg(myo6b:YFP-Hsa.TUBA)*$^{idc16Tg}$ larvae, while LysoTracker Green DND-26 (ThermoFisher, L7526) was used to label *Tg(myo6b:ctbp2-TagRFP)*$^{idc11Tg}$ larvae. Transgenic larvae were incubated in 100 nM LysoTracker dye in E3 media for 15 min and mounted in 1% low melt agarose prepared in E3 media containing 0.03% Tricaine-S and 100 nM LysoTracker dye.

To image LysoTracker label, samples were imaged live on a Nikon A1R upright confocal microscope using a $60\times1$ NA water objective lens. Denoised images were acquired using NIS Elements AR 5.20.02 with an 0.425 μm z-interval. Z-stacks of whole or partial neuromasts were acquired in a top-down configuration using 488 and 561 nm lasers.

## Immunohistochemistry

Immunohistochemistry was performed on whole larvae. Zebrafish larvae were fixed with 4% paraformaldehyde (Thermo Scientific, 28 906) in PBS for 3–4 h at 4°C. After fixation, samples were washed 5 × 5 min in PBS + 0.01% Tween (PBST), followed by a 5 min wash in H$_2$O. Larvae were then permeabilized with ice cold acetone (at −20 for 5 min. Larvae were then washed again in H$_2$O for 5 min, followed by 5 × 5 min washes in PBST, and then blocked overnight at 4°C with PBST containing 2% goat serum, 2% fish skin gelatin and 1% bovine serum albumin (BSA). Primary antibodies (Table 1) were diluted in PBST containing 1% BSA. Larvae were incubated in primary antibodies overnight at 4°C. After 5 × 5 min washes in PBST to remove the primary antibodies, larvae were incubated in diluted secondary antibodies (Table 2)

**Table 1. Primary antibody list**

| Antigen | Species; subtype | Concentration | Catalogue or reference | RRID/ZFIN ID |
|---|---|---|---|---|
| Pan-MAGUK | Mouse; IgG1 | 1:500 | Millipore MABN7 | RRID:AB_10807829 |
| Myo7A | Rabbit | 1:1000 | Proteus 25−6790 | RRID:AB_10015251 |
| Vglut3 C-term | Rabbit | 1:1000 | Obholzer et al., 2008 | ZDB-ATB-091117-1 |
| $Ca_V 1.3$ | Rabbit | 1:500 | Sheets et al., 2011 | ZDB-ATB-090515-1 |
| Ribeye b | Mouse; IgG2a | 1:10,000 | Sheets et al., 2011 | ZDB-ATB-120504-4 |
| Pan-CTBP | Mouse; IgG2a | 1:1000 | Santa Cruz 55502 | RRID:AB_629339 |
| Rab3a | Mouse; IgG1 | 1:1000 | Synaptic System 107 011 | RRID:AB_887768 |
| Acetylated tubulin | Mouse; IgG1 | 1:5000 | Sigma T7451 | RRID:AB_609894 |
| CSP | Rabbit | 1:1000 | Millipore AB1576 | RRID:AB_90794 |
| GFP | Chicken | 1:1000 | Thermofisher A10262 | RRID:AB_2534023 |

**Table 2. Secondary antibody list**

| Antigen | Species; fluorophore | Concentration | Catalogue or reference | RRID |
|---|---|---|---|---|
| Mouse IgG2a | Goat; Alexa 555, Alexa 633 | 1:1000 | ThermoFisher A-21137, A-21241 | RRID:AB_2535776, RRID:AB_2535810 |
| Mouse IgG1 | Goat; Alexa 488, Alexa 633 | 1:1000 | ThermoFisher A-21121, A-21240 | RRID:AB_2535764, RRID:AB_2535809 |
| Rabbit IgG | Goat; Alexa 488, Alexa 555, Alexa 633 | 1:1000 | ThermoFisher A-11008, A-21428, A-21071 | RRID:AB_143165, RRID:AB_141784 RRID:AB_141419 |
| Chicken IgY | Goat; Alexa 488 | 1:1000 | A-11039 | RRID:AB_2534096 |

(1:1000) in PBST containing 1% BSA for 3 h at room temperature. After 5 × 5 min washes in PBST to remove the secondary antibodies, larvae were rinsed in $H_2O$ and mounted in ProLong Gold Antifade (ThermoFisher, P36930).

### RNA-FISH to detect *kif1aa* and *kif1ab* mRNA in hair cells

To detect mRNA for *kif1aa and kif1ab*, we followed the Molecular Instrument RNA-FISH Zebrafish protocol, Revision Number 10 (https://files.molecularinstruments. com/MI-Protocol-RNAFISH-Zebrafish-Rev10.pdf), with a few minor changes to the preparation of fixed whole-mount larvae as follows. For our dehydration steps, we dehydrated using the following methanol series: 25, 50, 75, 100, 100% methanol, with 5 min for each step in the series. To permeabilize, we treated larvae with 10 μg/ml proteinase K for 20 min. RNA-FISH probes were designed to bind after the motor domain of zebrafish *kif1aa and kif1ab* (Fig. 1*F*, Molecular Instrument Probe lot # RTD364, RTD365, using B2 and B3 amplifiers, respectively). *Tg(myo6b:Cr.ChR2-EYFP)*[ahc1Tg] larvae were labelled using these probes; the strong EYFP label is retained after RNA-FISH and allows for delineation

of hair cells within the whole-mount larvae. After the RNA-FISH protocol, the larvae were mounted in ProLong Gold Antifade (ThermoFisher, P36930).

### Confocal imaging of RNA-FISH and immunolabels

Fixed immunostained and RNA-FISH samples were imaged on an inverted LSM 780 laser-scanning confocal microscope with an Airyscan attachment using Zen Blue 3.4 (Carl Zeiss) and an $63 \times 1.4$ NA Plan Apo oil immersion objective lens. Neuromast and inner ear z-stacks were acquired every 0.15 μm. The Airyscan z-stacks were auto-processed in 3D. Experiments were imaged with the same acquisition settings to maintain consistency between comparisons. For presentation in figures, images were further processed using Fiji (RRID:SCR_0 02285) (Schindelin et al., 2012).

### Quantification of RNA-FISH and synaptic components

Z-stack image acquisitions from zebrafish confocal images were processed in Fiji. Researchers were blinded to genotype during analyses. Hair-cell numbers were counted manually based on Myo7a, Cr.ChR2-EYFP, YFP-Hsa.TUBA or acetylated tubulin label. Each channel

was background subtracted using the rolling-ball radius method. Then each z-stack was max-intensity projected. A mask was generated by manually outlining the region or interest in the reference channel (e.g. hair cells via Myo7a, acetylated tubulin or *Tg(myo6b:Cr.ChR2-EYFP)*). This mask was then applied to the z-projection of each synaptic component or RNA-FISH channel.

We then used automated quantification to quantify puncta using a customized Fiji-based macro previously described (Jukic et al., 2024). In this macro, each masked image was thresholded using an adaptive thresholding plugin by Qingzong TSENG (https://sites.google.com/site/qingzongtseng/adaptivethreshold) to generate a binary image of the puncta (presynaptic, postsynaptic, $Ca_V1.3$ cluster or RNA-FISH puncta). Individual synaptic or RNA-FISH puncta were then segmented using the particle analysis function in Fiji. A watershed was applied to the particle analysis result to break apart overlapping puncta. After the watershed, the particle analysis was rerun with size and circularity thresholds to generate regions of interest (ROIs) and measurements of each punctum. For particle analysis, the minimum size thresholds of 0.025 $\mu m^2$ (Rib b, $Ca_V1.3$, *kif1aa*, and *kif1ab* RNA-FISH particles) and 0.04 $\mu m^2$ (MAGUK) were applied. A circularity factor of 0.1 was also used for the particle analysis. The new ROIs were applied to the original z-projection to get the average intensity and area of each punctum.

To identify paired synaptic components, images were further processed. Here, the overlap and proximity of ROIs from different channels (e.g. pre- and post-synaptic puncta) were calculated. ROIs with positive overlap or ROIs within two pixels were counted as paired components. The ROIs and synaptic component measurements (average intensity, area) and pairing results were then saved as Fiji ROIs, jpg images and csv files.

### Synaptic-vesicle quantification of live and fixed labels

Z-stack image acquisitions from live or fixed confocal images were processed and quantified in Fiji. A minimum of 3–6 hair cells per neuromast were analysed. Hair cells with a clear side view were used for base-to-apex analyses. For base-to-apex analyses, two rectangular ROIs with an area of 2.25 × 0.65 $\mu m$ (L × W) were placed above and below the nucleus of each hair cell, in one z-slice. Nuclei were identified based on acetylated tubulin or YFP-Hsa.TUBA labelling. The mean values were measured in all ROIs. The ratio of the base-to-apex was measured per hair cell to assess the enrichment of synaptic vesicles. Ratio values for hair cells were then averaged for a single neuromast base-to-apex fluorescence value. To quantify LysoTracker fluorescence at ribbons, a circular ROI was drawn around the ribbon indicated by Ctbp2a-tagRFP fluorescence and the mean value was measured in Fiji. In the cristae, Vglut3 label enrichment at the cell base was measured in the same manner as for neuromasts, but Vglut3 enrichment was only quantified in tall cells. To quantify the Vglut3 label in the anterior macula, a maximum projection of the stack imaged was generated in Fiji. An ROI was drawn outlining the macula, and the mean intensity values were measured.

### Startle behaviour

A Zantiks MWP behavioural system (https://zantiks.com) was used to assess acoustic startle responses in larvae at 5 dpf as previously described (Giese et al., 2023; Jukic et al., 2024). The Zantiks system tracks and monitors behavioural responses using an infrared camera at 30 frames per second. During the tracking and stimulation, a Cisco router connected to the Zantiks system was used to relay *x* and *y* coordinates of each larva in every frame. A 12-well plate was used for behavioural analyses. Each well was filled with E3 and one larva. All fish were acclimated in the plate within the Zantiks chamber in the dark for 15 min before each test. A vibrational stimulus that triggered a maximal proportion of animals startling in control animals without any tracking artefacts (due to the vibration) was used for our strongest stimuli. We used four different levels of intensity (1–4, increasing in intensity), with level 4 as the highest intensity stimulus. Based on previous literature, vibrations between 100 and 1000 Hz elicit short latency startle responses in zebrafish larvae (Beppi et al., 2021; Burgess & Granato, 2007). To deliver the acoustic-vibrational stimulus, the solenoid motor in the Zantiks system was set to move by 7.2° (level 4: 4 full steps), 3.6° (level 3: 2 full steps), 1.8° (level 2: 1 full step) and 0.9° (level 1: 1/2 step), with a 4 × 4.25 ms motor speed moving in clockwise and anticlockwise movements. We used an Optimus+ Red Sound Level Meter (Cirrus Research) to measure the intensity (dB) of each stimulus in the Zantiks chamber. The meter recorded the following sound intensities: 26.4 dB (Level 4), 23.3 dB (Level 3), 17.8 dB (Level 2) and 11.9 dB (Level 1). We chose these stimulus intensities based on the percentage of wild-type animals responding at each level. Level 4: 100%; level 3: 80%; level 2: 50%; level 1: 30%. Importantly, we have used this apparatus (at level 3) to parse out zebrafish with complete and moderate defects in the acoustic startle response (Giese et al., 2023). For our initial startle assay, each larva was presented with stimuli from intensity levels 1–3, five times, with 100 s between trials to avoid habituation. For each animal, the proportion of startle responses out of the five trials was plotted. For our habituation and recovery assay, a non-habituating stimulus, followed by a habituating stimulus train, and lastly a recovery stimulus train was performed as previously described (Marsden & Granato, 2015). Similar to previous studies, our non-habituating

stimulus was presented three times (intensity level 4) with 100 s between trials. This was followed by a habituating train of 30 stimuli (same stimulus intensity), presented with a 5 s inter-stimulus interval (ISI), an ISI shown to result in habituation (Marsden & Granato, 2015). We then presented recovery stimuli once each at 20 s, 40 s, 1 min and 2 min after the last stimulus in the habituating train. The proportion of startle responses out of the initial three non-habituating stimuli and the proportion of responses at each habituation block and recovery stimulus were plotted. To qualify as a startle response, a distance above four pixels, or ∼1.9 mm, was required within two frames after stimulus onset. Larvae were excluded from our analysis if no tracking data were recorded. Startle behaviour was performed on at least three independent days.

## Rheotaxis behaviour

A custom microflume (previously described in Newton et al., 2023) was used as the experimental apparatus for rheotaxis behaviour. Laminar water flow of a constant velocity was provided by a 6 V bow thruster motor (#108-01, Raboesch) inserted into the flume. An Arduino (UNO R3, Osepp) was programmed with custom scripts to coordinate the timing of the flow and video recording. An array of 196 LEDs emitting infrared light (850 nm) provided illumination for video capture through a layer of diffusion material (several Kimwipes sealed in plastic) and the translucent bottom of the flume. A monochromatic high-speed camera (SC1 without infrared filter, Edgertronic.com) with a 60 mm manual focus macro lens (Nikon) was used to record behavioural trials at 60 fps. The flume was filled with EM2 (28°C) and the arena was placed within the flume. Due to the heat generated by the infrared lights, the temperature was monitored, and miniature ice packs (2 × 2 cm; −20°C) were used to maintain a consistent temperature range of 27–29°C. For each rheotaxis behaviour test, individual 6 dpf larval zebrafish were transferred by pipette to the arena within the flume, and their swimming activity was monitored for ∼10 s to ensure that it exhibited burst-and-glide swimming behaviour; larva that did not exhibit normal swimming behaviour during the pre-trial period were not included in the analysis. Each larva was genotyped after behaviour acquisition and analysis. A total of 43 wild-type siblings and 30 *kif1aa* mutant larvae were analysed. Rheotaxis behaviour was assessed blind prior to genotyping. Data were collected from six experimental sessions on separate days.

Larval fish were tracked using DeepLabCut as previously described (Newton et al., 2023). In brief, video files were downsampled to 1000 × 1000 px and cropped. A previously created and trained single animal maDLC project was used to annotate seven unique body parts (left and right eyes, swim bladder, four points along the tail) on each larva. Videos were analysed with the maDLC project, and the detections were assembled into tracklets using the box method. The original videos were overlaid with the newly labelled body parts to check for tracklet accuracy. Misaligned tracklets were manually adjusted. Rheotaxis behaviour was annotated and analysed using a previously created custom Python feature extraction script (*SimBA*) that defined positive rheotaxis events as when the larvae swam into the oncoming water flow at an angle of 0° ± 45° for at least 100 ms (Goodwin et al., 2024). Videos processed through DeepLabCut analysis were converted to AVI format using the SimBA video editor function and imported into SimBA as previously described.

## Calcium imaging and electrophysiology

Larvae for electrophysiology recordings were either in a *Tg(myo6b:Cr.ChR2-EYFP)* transgenic background or a non-transgenic background. For calcium imaging, either *Tg(myo6b:memGCaMP6s)^{idc1Tg}* or *Tg(en.sill,hsp70l:GCaMP6s)^{idc8Tg}* transgenic larvae were used. To prepare larvae for calcium imaging and electrophysiology, 3–6 dpf larvae were anaesthetized in 0.03% Tricaine-S (SYNCAINE/MS-222, Syndel), pinned to a Sylgard-filled perfusion chamber at the head and tail. Then larvae were paralysed by injection of 125 μM $\alpha$-bungarotoxin (Tocris, 2133) into the heart cavity, as previously described (Lukasz & Kindt, 2018). Larvae were then rinsed once in E3 embryo media to remove the tricaine. Next, larvae were rinsed three times with extracellular imaging solution (in mM: 140 NaCl, 2 KCl, 2 CaCl$_2$, 1 MgCl$_2$ and 10 HEPES, pH 7.3, OSM 310 ± 10) and allowed to recover prior to calcium imaging or electrophysiology. Researchers were blind to genotype during the acquisition.

Calcium responses in the hair cells and afferent process were acquired on a Swept-field confocal system built on a Nikon FN1 upright microscope (Bruker) with a 60 × 1.0 NA CFI Fluor water-immersion objective. The microscope was equipped with a Rolera EM-C2 EMCCD camera (QImaging), controlled using Prairie view 5.4 (Bruker). GCaMP6s was excited using a 488 nm solid state laser. We used a dual band-pass 488/561 nm filter set (59904-ET, Chroma). For evoked measurements, stimulation was achieved by a fluid jet, which consisted of a pressure clamp (HSPC-1, ALA Scientific) and a glass pipette (pulled and broken to achieve an inner diameter of ∼50 μm) filled with extracellular imaging solution. A 500 ms pulse of positive or negative pressure was used to deflect the hair bundles of mechanosensitive hair cells along the anterior-posterior axis of the fish, as previously described (Giese et al., 2023). This stimulus duration is sufficient to parse out functional differences in the zebrafish lateral line (Kindig et al., 2023). For

GCaMP6s imaging in hair bundles or at the presynapse, the *Tg(myo6b:memGCaMP6s)^{idc1Tg}* line was used. For GCaMP6s imaging in the afferent terminal beneath lateral-line hair cells, the *Tg(en.sill,hsp70l:GCaMP6s)^{idc8Tg}* line was used. GCaMP6s measurements were performed on larvae at 4 and 5 dpf. Each neuromast (L1–L4) was stimulated four times with an ISI of ∼2 min. To acquire GCaMP6s evoked responses, five z-slices (0.5 µm step for mechanosensation, 1.5 µm step for presynaptic, and 2.0 µm step for the afferent process) were collected per timepoint for 80 timepoints at a frame rate of 20 ms for a total of ∼100 ms per z-stack and a total acquisition time of ∼8 s. Stimulation began at timepoint 31; timing of the stimulus was triggered by an outgoing voltage signal from Prairie view.

GCaMP6s z-stacks were average projected, registered and spatially smoothed with a Gaussian filter (size = 3, sigma = 2) in custom-written MatLab software as described previously (Zhang et al., 2018). The first 10 timepoints (∼1 s) were removed to reduce the effect of initial photobleaching. Registered average projections were analysed in Fiji to make intensity measurements using the Time Series Analyzer V3 plugin. Here, circular ROIs were placed on hair bundles or synaptic sites; average intensity measurements over time were measured for each ROI, as described previously (Lukasz & Kindt, 2018). GCaMP6s data were excluded in the case of excessive motion artefacts. Presynaptic responses were defined as >20% $\Delta$F/F0. Hair-bundle responses were defined as >20% $\Delta$F/F0. Postsynaptic responses were defined as >5% $\Delta$F/F0 and a minimum duration of 500 ms. Calcium imaging data then plotted in Prism 10 (Graphpad). The first 20 timepoints were averaged to generate an F0 value, and all responses were calculated as $\Delta$F/F0. The area under the curve function of Prism was used to determine the peak value for each response. Responses presented in figures represent average responses within a neuromast. The max $\Delta$F/F0 was compared between sibling and *kif1aa* mutants.

Extracellular postsynaptic current recordings from afferent cell bodies of the posterior lateral-line ganglion (pLLg) of zebrafish at 3–6 dpf were performed as previously described (Lukasz et al., 2022; Trapani & Nicolson, 2011). Briefly, borosilicate glass pipettes (Sutter Instruments, BF150-86-10 glass with filament) were pulled with a long taper, with resistances between 5 and 15 MΩ. The pLLg was visualized using an Olympus BX51WI fixed-stage microscope equipped with a LumPlanFl/IR 60 × 1.4 NA water-dipping objective (N2667800, Olympus). An Axopatch 200B amplifier, a Digidata 1400A data acquisition system and pClamp 10 software (Molecular Devices, LLC) were used to collect signals. To record spontaneous extracellular currents, afferent cell bodies were recorded using a loose-patch configuration with seal resistances ranging from 20 to 80 MΩ. Recordings were done in voltage-clamp mode, and signals were sampled at 50 µs/point and filtered at 1 kHz. The number of spontaneous events from one neuron per minute was quantified from a 5 min recording window using Igor Pro (Wavemetrics).

## Transmission electron microscopy

Larvae were genotyped at 2 dpf using a larval fin clip method to identify *kif1aa* and wild-type siblings to prepare for transmission electron microscopy (TEM) (Wilkinson et al., 2013). Larval tail DNA was genotyped as described above. At 5 dpf, *kif1aa* and wild-type siblings were fixed in a freshly prepared solution containing 1.6% paraformaldehyde and 2.5% glutaraldehyde in 0.1 M cacodylate buffer supplemented with 3.4% sucrose and 2 µM CaCl$_2$ for 2 h at room temperature, followed by a 24 h incubation at 4°C in a fresh portion of the same fixative. After fixation, larvae were washed with 0.1 M cacodylate buffer with supplements, and post-fixed in 1% osmium tetroxide for 30 min, and then washed with distilled water. Larvae were dehydrated in a 30–100% ethanol series, which included overnight incubation in 70% ethanol containing 2% uranyl acetate and in propylene oxide, and then embedded in Epon. Transverse serial sections (60–70 nm thin sections) were used to section through neuromasts. Sections were placed on single slot grids coated with carbon and formvar, and then sections were stained with uranyl acetate and lead citrate. All reagents and supplies for TEM were purchased from Electron Microscopy Sciences. Samples were imaged on a JEOL JEM-2100 electron microscope (JEOL Inc.). Whenever possible, serial sections were used to restrict our analysis to central sections of ribbons adjacent to the plasma membrane and a well-defined postsynaptic density.

To quantify ribbon area, ROIs were drawn in Fiji outlining the electron-dense ribbon, excluding the filamentous 'halo' surrounding the ribbon. Vesicles with a diameter of 30–50 nm and adjacent (within 60 nm of the ribbon) to the 'halo' were counted as tethered vesicles. Readily releasable vesicles were defined as tethered vesicles between the ribbon and the plasma membrane. To quantify reserve vesicles, we counted vesicles that were not tethered to the ribbon but were within 200 nm of the edge of the ribbon. All distances and perimeters were measured in Fiji.

## Statistics

All data shown are means and standard deviations unless stated otherwise. All replicates were biologically distinct animals and cells. Wild-type zebrafish animals were selected at random for drug treatments. *Kif1aa* mutants were compared with sibling controls, which were

**Table 3. Generalized Linear Mixed Model (GLMM) with Satterthwaite's method of *post hoc t* tests for differences in the total distance travelled during rheotaxis events. Type III ANOVA yielded significance values for fixed effects because the LME4 package in R does not identify them in its output. Significance codes: *** 0.001, * 0.05**

| GLMM | Estimated | Std. Error | Df | *t value* | Pr($>$|t|) |
|---|---|---|---|---|---|
| (Intercept) | 8.37 | 3.51 | 210 | 2.38 | **1.81E-02*** |
| Stimulus 10 s: Genotype mutant -/- | −9.25 | 7.46 | 141 | −1.24 | 0.217 |
| Stimulus 20 s: Genotype mutant -/- | −18.1 | 7.5 | 142 | −2.42 | **1.70E-02*** |

| ANOVA (III) | Sum Sq | Mean Sq | NumDF | DenDF | F value | Pr($>$F) |
|---|---|---|---|---|---|---|
| Stimulus | 46,700 | 23,400 | 2 | 142 | 47.5 | $< 2e-16$*** |
| Genotype | 1510 | 1510 | 1 | 71.1 | 3.07 | 0.0841 |
| Stimulus:genotype | 2870 | 1440 | 2 | 142 | 2.92 | 0.0573 |

**Table 4. Generalized Linear Mixed Model (GLMM) with Satterthwaite's method of *post hoc t* tests for differences in the mean number of rheotaxis events. Type III ANOVA yielded significance values for fixed effects because the LME4 package in R does not identify them in its output. Significance codes: *** 0.001, * 0.05**

| GLMM | Estimated | Std. Error | Df | *t value* | Pr($>$|t|) |
|---|---|---|---|---|---|
| (Intercept) | 0.0930 | 0.273 | 205 | 0.340 | 0.734 |
| Stimulus 10 s: Genotype mutant -/- | −0.350 | 0.558 | 142 | −0.627 | 0.532 |
| Stimulus 20 s: Genotype mutant -/- | −1.01 | 0.558 | 142 | −1.81 | 0.0728 |

| ANOVA (III) | Sum Sq | Mean Sq | NumDF | DenDF | F value | Pr($>$F) |
|---|---|---|---|---|---|---|
| Stimulus | 593 | 296 | 2 | 142 | 108 | $<2e-16$*** |
| Genotype | 2.74 | 2.74 | 1 | 71 | 0.997 | 0.321 |
| Stimulus:Genotype | 9.27 | 4.63 | 2 | 142 | 0.189 | 0.189 |

either wild-type siblings or a combination of wild-type and *kif1aa* heterozygous siblings obtained from the same clutch. All experiments were performed blind to genotype. In all datasets, dot plots represent the '*n*'. N represents either the number of neuromasts, hair cells, synapses or puncta as stated in the legends. All experiments were replicated on multiple independent days. For our experiments, a minimum of three animals and eight neuromasts were examined. An exception is our TEM experiments, where we examined ribbon synapses in seven neuromasts from four wild-type siblings and four neuromasts from three *kif1aa* mutants. All animal counts are listed in Table 5. Sample sizes were selected to avoid type 2 error. Statistical analyses were performed using Prism 10 software (GraphPad) or in R for our rheotaxis assays. A D'Agostino-Pearson normality test was used to test for normal distributions. To test for statistical significance between two samples with normal distributions, an unpaired *t* test was used. For acoustic startle assays, a two-way ANOVA with multiple comparisons was used. A Sidak test was used to correct for multiple comparisons. For rheotaxis behaviour: data wrangling, cleaning and figure generation were performed with R and performed as previously described (Newton et al., 2022). The mean duration, number, total distance travelled and mean latency to the onset of rheotaxis events were calculated for 10 s bins (pre-stimulus, first 10 s of stimulus and last 10 s of stimulus) using SimBA and analysed in R using generalized linear mixed models (GLMM) and *post hoc t* tests (Satterthwaite method). Type III ANOVAs provided significance values for the fixed effects of the GLMMs (see Tables 3, 4). A *P* value less than 0.05 was considered significant. All error bars and ± are standard deviations, unless stated otherwise.

## Results

### *Kif1aa* is expressed in lateral-line hair cells and impacts ribbon-synapse counts

Work in neurons has demonstrated that KIF1A is an anterograde kinesin motor that transports synaptic-vesicle precursors towards the plus ends of microtubules (Okada et al., 1995). Recent work in both mouse and zebrafish hair cells has demonstrated that microtubules are polarized, with their plus ends growing

**Table 5. Animal counts for all experiments described in paper**

| Figure | Experiment | Number of animals | N represented in dot plots |
|---|---|---|---|
| In results | *kif1aa* RNA-FISH puncta per neuromast | sibling control 7; *kif1aa* 5 | sibling control 18 neuromasts; *kif1aa* 16 neuromasts |
| Fig. 3C–F | Ribbon synapse counts and area | sibling control 9; *kif1aa* 7 | sibling control 18 neuromasts; *kif1aa* 15 neuromasts |
| Fig. 4D | Lysotracker | sibling control 4; *kif1aa* 3 | sibling control 10 neuromasts; *kif1aa* 10 neuromasts |
| Fig. 4H | Vglut3 label | sibling control 4; *kif1aa* 5 | sibling control 12 neuromasts; *kif1aa* 13 neuromasts |
| Fig. 4J | CSP label | sibling control 5; *kif1aa* 8 | sibling control 10 neuromasts; *kif1aa* 10 neuromasts |
| Fig. 4L | Rab3a label | sibling control 4; *kif1aa* 3 | sibling control 9 neuromasts; *kif1aa* 9 neuromasts |
| Fig. 5C | Vglut3 crista | sibling control 8; *kif1aa* 7 | sibling control 8 crista; *kif1aa* 8 crista |
| Fig. 5F | Vglut3 anterior macula | sibling control 4; *kif1aa* 5 | sibling control 6 macula; *kif1aa 6* macula |
| Fig. 6D | Lysotracker | DMSO control 4; Nocodazole 3 | DMSO control 10 neuromasts; nocodazole 10 neuromasts |
| Fig. 6H | Vglut3 | DMSO control 3; Nocodazole 4 | DMSO control 8 neuromasts; nocodazole 10 neuromasts |
| Fig. 7C | Lysotracker on ribbons | sibling control 4; *kif1aa* 3 | sibling control 10 neuromasts; *kif1aa* 10 neuromasts |
| Fig. 7I–J | TEM RRP and docked vesicle | Wild-type control 4; *kif1aa* 3 | wild-type control 14 ribbons; *kif1aa* 14 ribbons |
| Fig. 7K | TEM reserve pool | wild-type control 4; *kif1aa* 3 | wild-type control 12 ribbons; *kif1aa* 13 ribbons |
| Fig. 8C–F | Cav1.3 immunolabel | sibling control 11; *kif1aa* 6 | sibling control 13 neuromasts; *kif1aa* 9 neuromasts |
| Fig. 9F–I | Mechanoelectrical transduction and presynapse responses | sibling control 7; *kif1aa* 6 | sibling control 8 neuromasts; *kif1aa* 8 neuromasts |
| Fig. 10C | Afferent spikes | sibling control 8; *kif1aa* 11 | sibling control 8 neurons; *kif1aa* 13 neurons |
| Fig. 10G–H | Afferent calcium responses | sibling control 8; *kif1aa* 5 | sibling control 10 neuromasts; *kif1aa* 9 neuromasts |
| Fig. 11A | Acoustic startle 3 stim intensities | sibling control 38; *kif1aa* 13 | sibling control 38 animals; *kif1aa* 13 animals |
| Fig. 11B | Acoustic startle habituation | sibling control 38; *kif1aa* 25 | sibling control 38 animals; *kif1aa* 25 animals |
| Fig. 11E–F | Rheotaxis | wild-type control 43; *kif1aa* 30 | wild-type control 43 animals; *kif1aa* 30 animals |

toward the cell base, near the presynaptic AZ (Fig. 1B; Hussain et al., 2024; Voorn et al., 2024). Further, RNAseq data indicate robust levels of *Kif1a* transcripts in mouse auditory hair cells and zebrafish hair cells (Kolla et al., 2020; Lush et al., 2019; Sur et al., 2023). Based on work in neurons, we used zebrafish to test whether Kif1a is a kinesin motor involved in the transport of synaptic vesicles to ribbon synapses in hair cells.

In zebrafish, there are two paralogues of mammalian *Kif1a*, *kif1aa* and *kif1ab* (Fig. 1F). Kif1aa and Kif1ab proteins show a high degree of identity to each other

(89%, NCBI BLAST) and to human KIF1A (84% and 85% identity, respectively, NCBI BLAST). Recent scRNAseq work has detected widespread expression of both *kif1aa* and *kif1ab* transcripts throughout the zebrafish nervous system (Sur et al., 2023). Similarly, widespread expression of *Kif1a* in the nervous system of mice is also well documented (Okada et al., 1995). We used high-resolution RNA fluorescence *in situ* hybridization (RNA-FISH) to verify whether the *kif1aa* and *kif1ab* mRNAs are present in zebrafish hair cells (Choi et al., 2018) and found that only *kif1aa* mRNA was present in hair cells of the

zebrafish lateral line (Fig. 2*A* and *B*). In contrast, both *kif1aa* and *kif1ab* mRNA were present in the hair cells of the zebrafish inner ear (Fig. 2*C–F*). In the inner ear, we observed both *kif1aa* and *kif1ab* mRNA in the cristae (which detects angular acceleration), the anterior macula (which detects gravity and is required for balance) and the posterior macula (which primarily detects auditory stimuli) (Fay & Popper, 2000).

After validating that *Kif1a* ohnologues are indeed expressed in hair cells, we leveraged zebrafish genetics to test whether Kif1a plays a role in the transport of synaptic vesicles in hair cells. We focused our efforts on a germline *kif1aa* mutant we created in a previous study (Hussain et al., 2024). This mutant has a stop codon in the Kif1aa motor domain and is predicted to be a null mutation (Fig. 1*F*). This mutant should impair Kif1a in lateral-line hair cells but leave Kif1a function at least partially intact in the inner ear and the nervous system, due to the expression of the ohnologue *kif1ab*. We found via RNA-FISH that *kif1aa* mRNA expression was reduced by 60% in lateral-line hair cells in *kif1aa* mutants compared to sibling controls (control: 403 ± 65.8 puncta per neuromast; *kif1aa*: 173 ± 48.6 puncta per neuromast, *n* = 18 control and 16 *kif1aa* neuromasts, unpaired

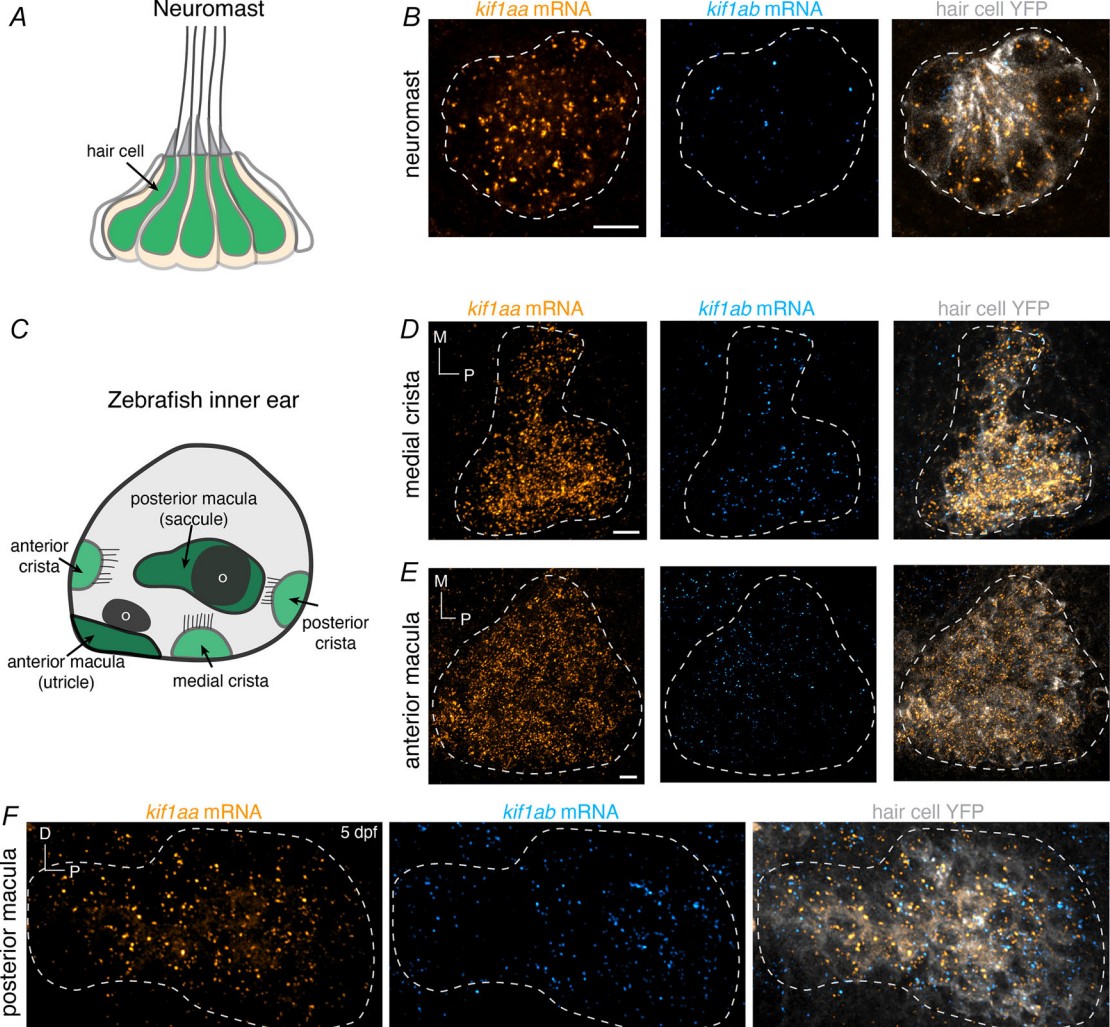

**Figure 2. *kif1aa* mRNA is present in neuromast hair cells, while both *kif1aa* and *kif1ab* mRNAs are present in inner-ear hair cells**
*A,C*, schematics showing a zebrafish neuromast and inner ear at 5 dpf. Within the inner ear, hair cells are present in three cristae and two maculae. Each macula is associated with an otolith (o). *B*, RNA-FISH shows that at 5 dpf only *kif1aa* (orange) mRNAs are present in neuromast hair cells. *D–F*, in the inner ear, RNA-FISH shows that at 5 dpf, both *kif1aa* (orange) and *kif1ab* (cyan) mRNAs are present in hair cells in cristae and maculae. The grey label is YFP, which is expressed specifically in hair cells by the transgenic line *Tg(myo6b:Cr.ChR2-EYFP)*. The YFP label was used to create the dashed line in *B, D, E, F* to outline the locations of hair cells within each sensory epithelium. Scale bars = 5 μm.

*t* test, *P* < 0.0001). Overall, this reduction in mRNA levels provides evidence that we have disrupted the *kif1aa* locus and that there is significant nonsense-mediated degradation of *kif1aa* mRNA in the lateral-line hair cells of *kif1aa* mutants.

Before assessing synaptic vesicles in *kif1aa* mutant hair cells, we first performed a gross assessment of lateral-line neuromasts using immunohistochemistry to label hair cells and ribbon synapses (Fig. 3*A*–*B*). We performed our analyses at 5 dpf, when the lateral-line system is functional (Suli et al., 2012). We used antibodies against Myosin7a (Myo7a) to label hair cells, along with Ribeye b (Rib b) and membrane-associated guanylate kinase (MAGUK) to label the pre- and postsynapses, respectively. We found that *kif1aa* mutants have a similar number of hair cells per neuromast compared with sibling controls (Fig. 3*C*, control: 15.6 ± 1.3; *kif1aa*: 16.0 ± 1.7, *n* = 18 control and 15 *kif1aa* neuromasts; unpaired *t* test, *P* = 0.399). For our synapse analysis, we examined 2D maximum intensity projections and quantified the number and

area of ribbons and postsynapses. We found that there were significantly fewer complete synapses (paired Rib b-MAGUK puncta) per hair cell in mature lateral-line hair cells (5 dpf) in *kif1aa* mutants compared with sibling controls (Fig. 3*D*, control: 3.13 ± 0.35; *kif1aa*: 2.68 ± 0.14, *n* = 18 control and *kif1aa* 15 neuromasts; unpaired *t* test, *P* = 0.00870). Fewer synapses is consistent with our work which demonstrated that Kif1aa also plays a subtle role in presynapse assembly (Hussain et al., 2024). Quantification of pre- and post-synapse size at complete synapses revealed no difference in presynapse size but slightly larger postsynapses in *kif1aa* mutants compared with controls (Fig. 3*E*, presynapse area, control: 0.21 ± 0.02; *kif1aa*: 0.20 ± 0.02, unpaired *t* test, *P* = 0.345; Fig. 3*F*, post-synapse area, control: 0.19 ± 0.02; *kif1aa*: 0.20 ± 0.02, unpaired *t* test, *P* = 0.0498). Overall, we found that *Kif1a* paralogues are present in hair cells in zebrafish and that loss of Kif1aa in the lateral line does not alter hair-cell counts but does result in slightly fewer ribbon synapses.

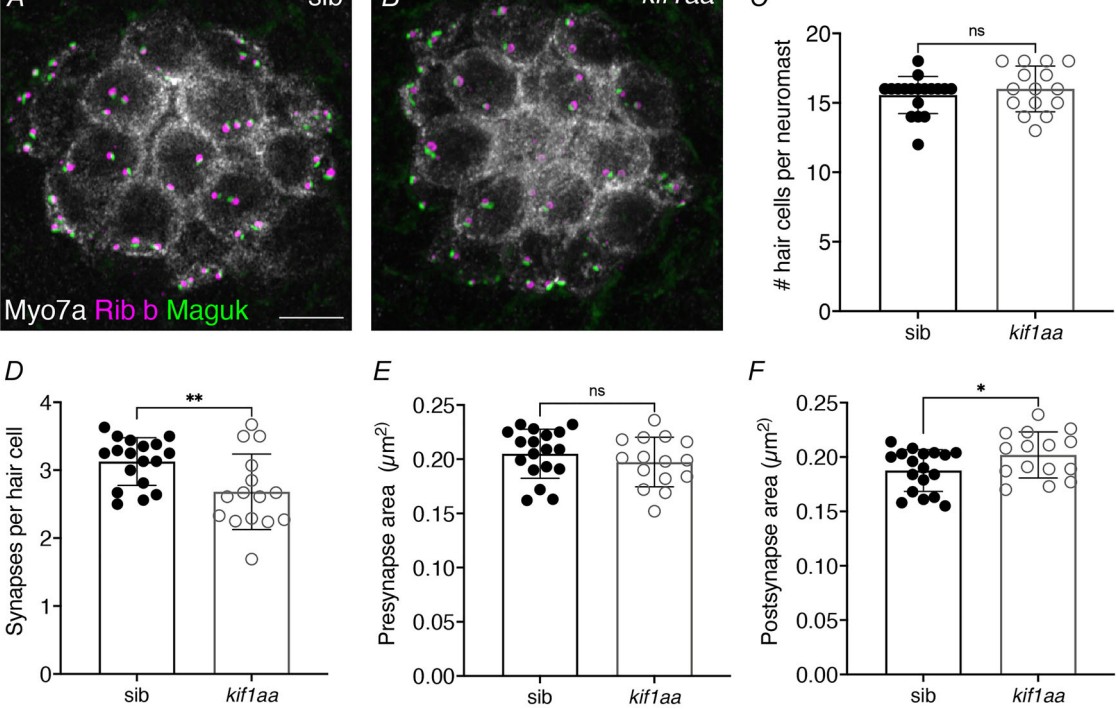

**Figure 3. *Kif1aa* mutants have fewer ribbon synapses**
*A,B*, example immunolabel of neuromasts at 5 dpf in a *kif1aa* mutant (*B*) or sibling control (*A*). Myosin7a (Myo7a) labels hair cells, Ribeye b (Rib b) labels ribbons or presynapses, and MAGUK labels postsynapses. *C*, the number of hair cells per neuromast is the same in *kif1aa* mutants compared with sibling controls (control: 15.6 ± 1.3; *kif1aa*: 16.0 ± 1.7, unpaired *t* test, *P* = 0.399). *D*, *Kif1aa* mutants have fewer complete synapses per cell (control: 3.13 ± 0.35; *kif1aa*: 2.68 ± 0.56, unpaired *t* test, *P* = 0.00870). *E*, in *kif1aa* mutants, the average area of Rib b puncta (presynaptic) was similar to sibling controls (control: 0.21 μm² ± 0.02; *kif1aa*: 0.20 μm² ± 0.02, unpaired *t* test, *P* = 0.345). *F*, in *kif1aa* mutants, the average area of MAGUK puncta (postsynaptic) was slightly larger compared with sibling controls (control: 0.19 μm² ± 0.02; *kif1aa*: 0.20 μm² ± 0.02, unpaired *t* test, *P* = 0.0498). *n* = 18 control and 15 *kif1aa* neuromasts in *C–F*. Scale bar in *A* = 5 μm.

## Kif1aa is essential for synaptic-vesicle distribution in hair cells

Our overall assessment suggests that in *kif1a* mutants, ribbon synapses are largely intact, although fewer in number. As previous work in neurons has shown that Kif1a transports synaptic-vesicle precursors to the pre-synaptic AZ, we assessed whether loss of Kif1aa impacts synaptic-vesicle distribution at the presynapse in hair cells. We imaged both live and fixed preparations to visualize synaptic-vesicle distributions in hair cells of *kif1aa* mutants. To label synaptic vesicles live, we incubated zebrafish for 15 min with 100 nM of the vital dye LysoTracker. Previous work used this labelling approach in zebrafish to label acidified organelles, including synaptic vesicles, in living lateral-line hair cells (Einhorn et al., 2012). To label synaptic vesicles in fixed samples, we used immunohistochemistry to label Vglut3 (vesicular glutamate transporter 3), a marker of synaptic vesicles in hair cells of zebrafish and mice (Obholzer et al., 2008; Schraven et al., 2012).

In line with previous work, we observed an enrichment of LysoTracker dye at the cell base near the presynaptic AZ in sibling controls (Fig. 4A and C; Einhorn et al., 2012). In contrast, we did not observe enrichment of LysoTracker at the base of hair cells in *kif1aa* mutants (Fig. 4B and C). To quantify LysoTracker enrichment, we measured the base-to-apex ratio of fluorescence in individual hair cells (Fig. 4C). Using this metric, we found that *kif1aa* mutants enrich significantly less LysoTracker at the cell base (Fig. 4D, control: $7.38 \pm 1.16$; *kif1aa*: $0.52 \pm 0.16$ $n = 10$ control and *kif1aa* neuromasts, unpaired *t* test, $P < 0.0001$). Interestingly, we observed that the base-to-apex ratio was less than one in Kif1aa-deficient hair cells, indicating that the LysoTracker label is more enriched in the hair-cell apex in *kif1aa* mutants.

Similar to the LysoTracker label and consistent with previous work in lateral-line hair cells, we observed that the Vglut3 immunolabel was enriched at the cell base in sibling controls (Fig. 4E and G; Obholzer et al., 2008). This enrichment was not observed in *kif1aa* mutants (Fig. 4F and G). We quantified enrichment by measuring the base-to-apex ratio of the Vglut3 immunolabel. We found that the base-to-apex ratio of the Vglut3 label was significantly reduced in *kif1aa* mutants compared with sibling controls (Fig. 4H, control: $2.24 \pm 0.65$; *kif1aa*: $0.45 \pm 0.07$; $n = 12$ control and 13 *kif1aa* neuromasts, unpaired *t* test, $P < 0.0001$).

To further verify that *kif1aa* mutants fail to enrich synaptic vesicles at the base of lateral-line hair cells, we used immunohistochemistry to label Rab3a and cysteine string protein (CSP) (Fig. 4I–L). Rab3a and CSP are synaptic-vesicle markers that have also been shown to be enriched at the base of lateral-line hair cells (Einhorn et al., 2012; Kindt & Sheets, 2018). We measured the base-to-apex ratio of the Rab3a and CSP immunolabels

to quantify synaptic-vesicle enrichment. We observed that similar to our Vglut3 immunolabel, the base-to-apex ratio of both Rab3a and CSP immunolabels was significantly reduced in *kif1aa* mutants compared with sibling controls (Fig. 4J, CSP, control: $1.86 \pm 0.34$; *kif1aa*: $0.58 \pm 0.07$; $n = 10$ control and *kif1aa* neuromasts, unpaired *t* test, $P < 0.0001$ and Fig. 4L, Rab3a, control: $1.57 \pm 0.30$; *kif1aa*: $0.63 \pm 0.15$; $n = 9$ control and *kif1aa* neuromasts, unpaired *t* test, $P < 0.0001$).

In addition to lateral-line hair cells, we also examined synaptic-vesicle distribution in inner-ear hair cells in *kif1aa* mutants. For our analysis, we focused on Vglut3 labelling, as Vglut3 has been shown to be essential for both inner ear and lateral-line function in zebrafish (Obholzer et al., 2008). First, we examined Vglut3 distribution in cristae. Interestingly, in sibling controls, we observed that the Vglut3 immunolabel was only enriched in a subset of the hair cells (Fig. 5A, blue outlines). Previous work has shown that zebrafish cristae have two main hair-cell types that can be distinguished morphologically as 'tall' and 'short' hair cells (Smith et al., 2020; Zhu et al., 2020). In sibling controls, we found that the tall cells enriched Vglut3 at the cell base. Compared with tall cells, short cells had nearly undetectable levels of Vglut3 (Fig. 5A, yellow outlines). Importantly, when we examined *kif1aa* mutants, we found that tall cells enriched significantly less Vglut3 immunolabel at the cell base (Fig. 5C, control: $3.38 \pm 0.86$, *kif1aa*: $1.35 \pm 0.29$, $n = 8$ control and *kif1aa* crista, unpaired *t* test, $P < 0.0001$). This observation indicates that Kif1aa is required for enrichment of Vglut3 at the base of tall cells within cristae.

In addition to cristae, we also examined the Vglut3 immunolabel in the anterior macula. Unlike in the cristae, we found that in sibling controls all macular hair cells had detectable amounts of Vglut3 immunolabel and showed enrichment of Vglut3 at the cell base (Fig. 5D–E). In addition, we found that Vglut3 immunolabel was still enriched at the cell base in the macular hair cells of *kif1aa* mutants (Fig. 5D–E). Due to the depth of this epithelium, we were unable to accurately quantify the extent of enrichment of Vglut3 at the base of macular hair cells. Instead, we quantified the mean intensity of Vglut3 immunolabel within the anterior macula as a whole. This quantification revealed that the mean intensity of Vglut3 was significantly lower in *kif1aa* mutants (Fig. 5F, control: $428 \pm 105$, *kif1aa*: $260 \pm 119$, $n = 6$ control and *kif1aa* anterior macula, unpaired *t* test, $P = 0.0264$). This indicates that Kif1aa impacts the amount of synaptic vesicles in hair cells of the anterior macula.

Overall, our live and fixed imaging of synaptic-vesicle distribution in *kif1aa* mutants indicates that Kif1aa is essential for enriching synaptic vesicles at the presynaptic AZ of lateral-line hair cells and specifically in tall cells of the cristae. In contrast, loss of Kif1aa has a more moderate impact on synaptic-vesicle distribution in macular hair cells.

## An intact microtubule network is essential for synaptic-vesicle distribution

KIF1A is known to transport synaptic-vesicle precursors towards the plus ends of microtubules. Therefore, we employed a pharmacological approach to understand the role that an intact microtubule network plays in synaptic-vesicle distribution in lateral-line hair cells. Nocodazole has been shown to block the self-assembly of tubulin and depolymerize preassembled microtubules (Samson et al., 1979). In recent work, we demonstrated that 30 min, 4 h and 16 h treatments with 250–500 nm

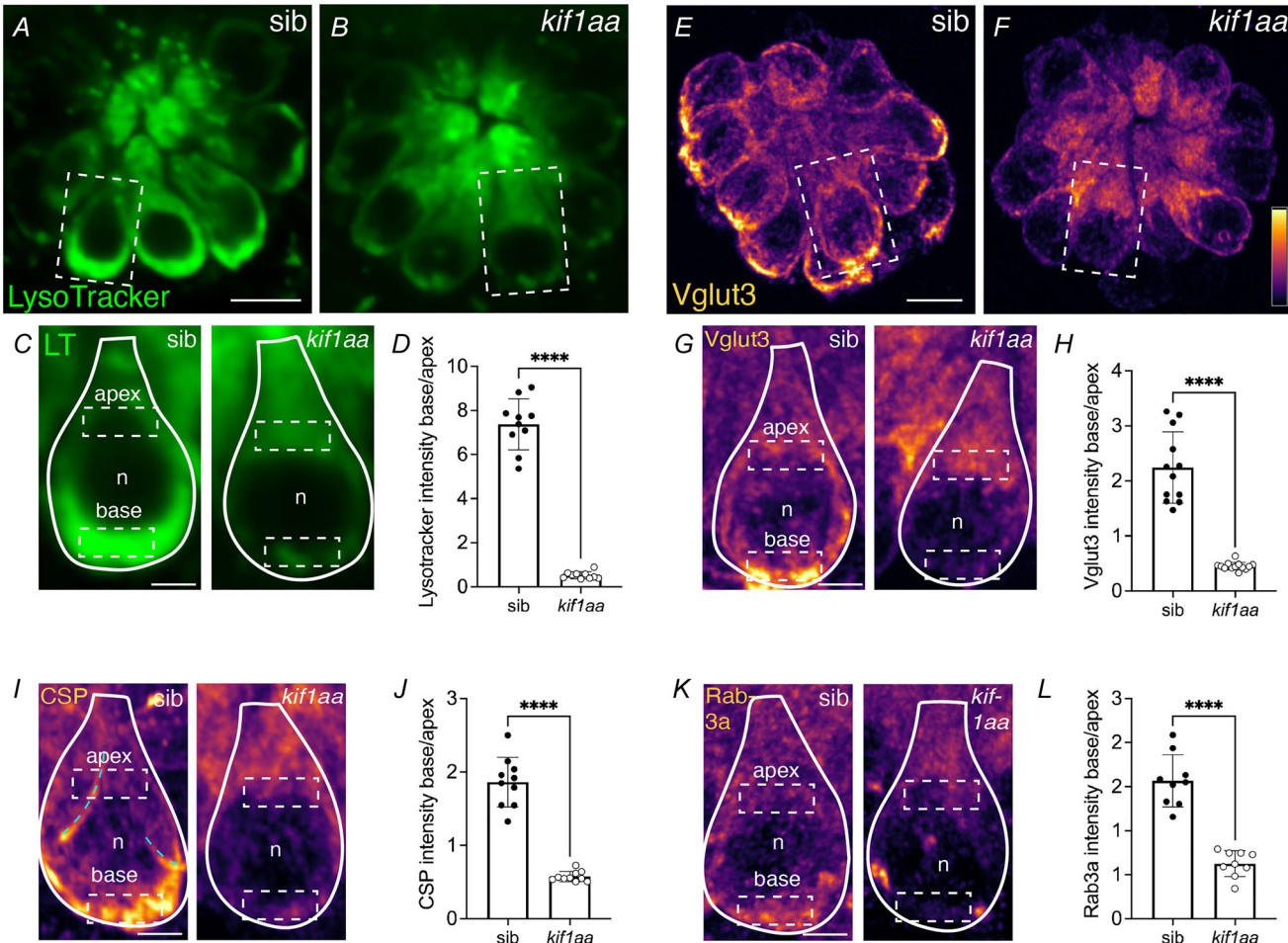

**Figure 4. *Kif1aa* mutants enrich less LysoTracker, Vglut3, CSP and Rab3a at the presynapse**
*A–C*, example live image of LysoTracker Red (green) to label synaptic vesicles in neuromasts at 5 dpf in a *kif1aa* mutant (*B*) or sibling control (*A*). The dashed box in each image indicates the hair cells magnified and outlined with a continuous line in *C*. The dashed box in each image indicates example ROIs of the apical and basal regions used for intensity analysis in *D*. *D*, quantification of LysoTracker shows that in *kif1aa* mutants there is significantly less enrichment at the cell base compared with sibling controls (control: 7.38 ± 1.16; *kif1aa*: 0.52 ± 0.16, *n* = 10 *kif1aa* and control neuromasts, unpaired *t* test, *P* < 0.0001). *E–G*, example immunostain of Vglut3 to label synaptic vesicles in neuromasts at 5 dpf in a *kif1aa* mutant (*F*) or sibling control (*E*). The dashed box in each image indicates the hair cells magnified and outlined with a continuous line in *G*. *H*, quantification reveals that Vglut3 is significantly less enriched at the cell base in *kif1aa* mutants compared with sibling controls (control: 2.24 ± 0.65; *kif1aa*: 0.45 ± 0.07, *n* = 12 control and 13 *kif1aa* neuromasts, unpaired *t* test, *P* < 0.0001). *I*, examples of neuromast hair cells immunolabeled with CSP. Cyan dashed lines indicate CSP label from neighbouring cells. *J*, quantification of CSP label reveals that in *kif1aa* mutants, there is significantly less CSP enriched at the cell base (control: 1.86 ± 0.34, *kif1aa*: 0.58 ± 0.07, *n* = 10 control and *kif1aa* neuromasts, unpaired *t* test, *P* < 0.0001). *K*, examples of neuromast hair cells immunolabeled with Rab3a. Residual basal puncta in *kif1aa* mutants are efferent terminals contacting hair cells that also have high levels of Rab3a. *L*, quantification reveals that the Rab3a label is significantly less enriched at the cell base (control: 1.57 ± 0.30, *kif1aa*: 0.63 ± 0.15, *n* = 9 control and *kif1aa* neuromasts, unpaired *t* test, *P* < 0.0001). The solid lines in the magnified images outline a single hair cell, with the base of the cell at the bottom of the image, and dashed boxes indicate example ROIs of the apical and basal regions used for quantification. n indicates nucleus. Scale bars in *A* and *E* = 5 µm and 2 µm in *C, G, I, K*.

nocodazole disrupt microtubule networks in lateral-line hair cells (Hussain et al., 2024). For our present study, we incubated larvae in 250 nᴍ nocodazole for 2 h to disrupt microtubules, a concentration and incubation duration that did not result in hair-cell death. We used nocodazole along with LysoTracker label or Vglut3 immunolabel to assess the impact of microtubule disruption on synaptic-vesicle distribution.

After incubation with 250 nᴍ nocodazole, we observed less basal enrichment of LysoTracker compared with DMSO controls (Fig. 6A–C). To quantify label enrichment, we measured the base-to-apex ratio of LysoTracker fluorescence in hair cells (Fig. 6C–D) and found significantly less basal enrichment of LysoTracker in nocodazole-treated hair cells compared with DMSO controls (Fig. 6D, control: 6.68 ± 1.43; nocodazole: 1.22 ± 0.37 *n* = 10 control and nocodazole neuromasts,

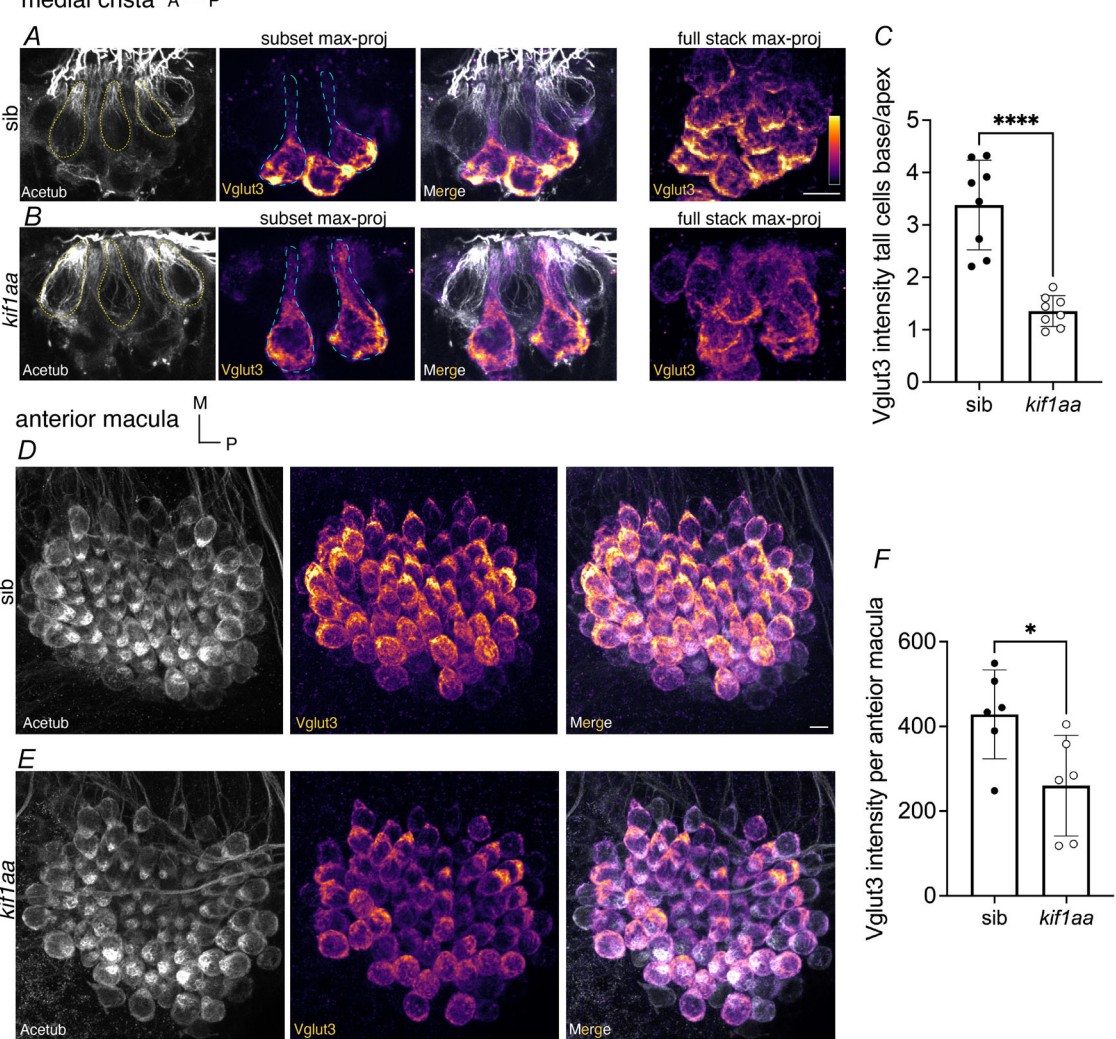

**Figure 5. *Kif1aa* mutants disrupt Vglut3 localization in subsets of inner-ear hair cells**
*A,B*, immunolabel of hair cells in the medial crista with acetylated tubulin (Acetub, grey) to mark hair cells and Vglut3 to mark synaptic vesicles in *kif1aa* mutants (*B*) and sibling controls (*A*) at 5 dpf. In the crista of both *kif1aa* mutants and controls, only a subset of hair cells (tall cells), show high levels of Vglut3 (cells outlined with cyan dashed lines in *A* and *B*). In contrast, other hair cells (tear drop cells) have no detectable Vglut3 (cells outlined with yellow dashed lines in *A* and *B*). *C*, quantification reveals that the Vglut3 label is significantly less enriched at the cell base of tall cells in the cristae of *kif1aa* mutants compared with sibling controls (control: 3.38 ± 0.86, *kif1aa*: 1.35 ± 0.29, *n* = 8 control and *kif1aa* cristae, unpaired *t* test, *P* < 0.0001) (see partial and full stack max-projected images in *A* and *B*). *D,E*, immunolabel of hair cells in the anterior macula with acetylated tubulin (Acetub, grey) to mark hair cells and Vglut3 to mark synaptic vesicles in *kif1aa* mutants (*D*) and sibling controls (*E*) at 5 dpf. *F*, in the anterior macula, the mean intensity of Vglut3 immunolabel in the maculae was significantly reduced in *kif1aa* mutants compared with control (control: 428 ± 105, *kif1aa*: 260 ± 119, *n* = 6 control and *kif1aa* anterior macula, unpaired *t* test, *P* = 0.0264). Scale bars in *A* and *D* = 5 μm.

unpaired *t* test, *P* < 0.0001). In addition, we found significantly less basal enrichment of Vglut3 in hair cells in nocodazole-treated larvae compared with DMSO controls (Fig. 6*E–H*, control: 5.75 ± 1.61; nocodazole: 2.53 ± 0.65; *n* = 8 control and nocodazole neuromasts, unpaired *t* test, *P* < 0.0001). Together, our nocodazole treatments indicate that both an intact microtubule network and functional Kif1aa are necessary to localize synaptic vesicles at the presynapse in lateral-line hair cells.

### Kif1aa is important to maintain synaptic vesicles at ribbon synapses

Our live and fixed confocal imaging strongly suggest that synaptic vesicles fail to enrich at the cell base in *kif1aa* mutants (Figs 4 and 5). Compared with sibling controls, the overall apex-to-base enrichment of Vglut3 and LysoTracker labels was reduced by 80% and 92%, respectively, in *kif1aa* mutants. To understand how synaptic vesicles are impacted more focally at sites of release, we examined synaptic vesicles at individual ribbons in lateral-line hair cells. We first examined our LysoTracker label more closely at individual ribbons. To detect ribbons within the LysoTracker label, we used a transgenic line that labels ribbons *Tg(myo6b:Rib*

*a-TagRFP)*. In addition to LysoTracker measurements, we also used TEM to visualize individual synaptic vesicles at ribbons.

In control hair cells, similar to previous observations in lateral-line hair cells, we observed an intense LysoTracker label or halo surrounding individual ribbons (Einhorn et al., 2012) (Fig. 7*A–B*). This halo represents, in part, the population of synaptic vesicles tethered near the ribbon. We quantified the amount of LysoTracker in ROIs directly on individual ribbons, measuring LysoTracker intensity in a single plane that encompassed the centre of the ribbon. Using this approach, we found a 30% reduction in LysoTracker label at individual ribbons in *kif1aa* mutants compared with sibling controls (Fig. 7*C*, control: 2580 ± 450; *kif1aa*: 1810 ± 650, *n* = 10 control and *kif1aa* neuromasts, unpaired *t* test, *P* = 0.0003). This indicates that there is less LysoTracker label and likely fewer synaptic vesicles near ribbons in *kif1aa* mutants.

To quantify synaptic vesicles more definitively, we used TEM to visualize ribbons in lateral-line hair cells as previously described (Sheets et al., 2017). We only examined micrographs that contained images of a centrally sectioned ribbon adjacent to a well-defined postsynaptic density (Fig. 7*D–H*). We quantified three main populations of synaptic vesicles localized in the vicinity

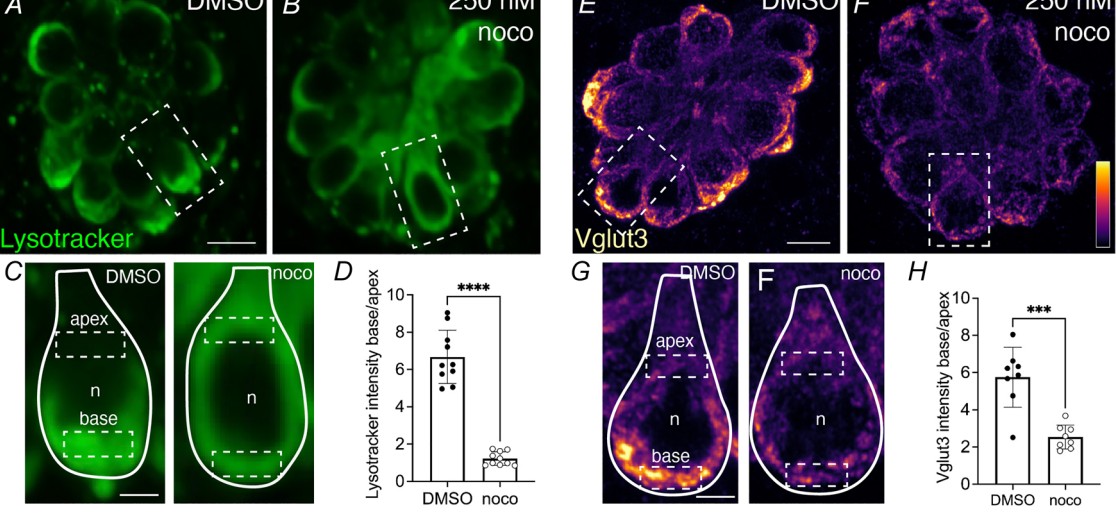

**Figure 6. An intact microtubule network is required to enrich LysoTracker and Vglut3 at the presynapse**
*A–C*, example live image of LysoTracker Red (green) to label synaptic vesicles in neuromasts at 5 dpf in wild-type larva treated with 250 nM nocodazole (*B*) or DMSO control (*A*). The dashed box in each image indicates the hair cell magnified and outlined with a continuous line in *C*. *D*, quantification shows significantly less LysoTracker enrichment at the cell base in nocodazole-treated larvae compared with DMSO control (control: 6.68 ± 1.43; nocodazole: 1.22 ± 0.37, *n* = 10 control and nocodazole neuromasts, unpaired *t* test, *P* < 0.0001). *E–G*, example immunolabel of Vglut3 to label synaptic vesicles in neuromasts at 5 dpf in larva treated with 250 nM nocodazole (*F*) or DMSO control (*E*). The dashed box in each image indicates the hair cell magnified and outlined with a continuous line in *G*. *H*, quantification reveals significantly less Vglut3 enrichment at the cell base in nocodazole-treated larvae compared with DMSO controls (control: 5.75 ± 1.61; nocodazole: 2.53 ± 0.65, *n* = 8 control and nocodazole neuromasts, unpaired *t* test, *P* < 0.0001). The solid lines in the magnified images outline a single hair cell, with the base of the cell at the bottom of the image, and dashed boxes indicate example ROIs of the apical and basal regions used for quantification. n indicates nucleus. Scale bars in *A* and *E* = 5 μm and 2 μm in *C* and *G*.

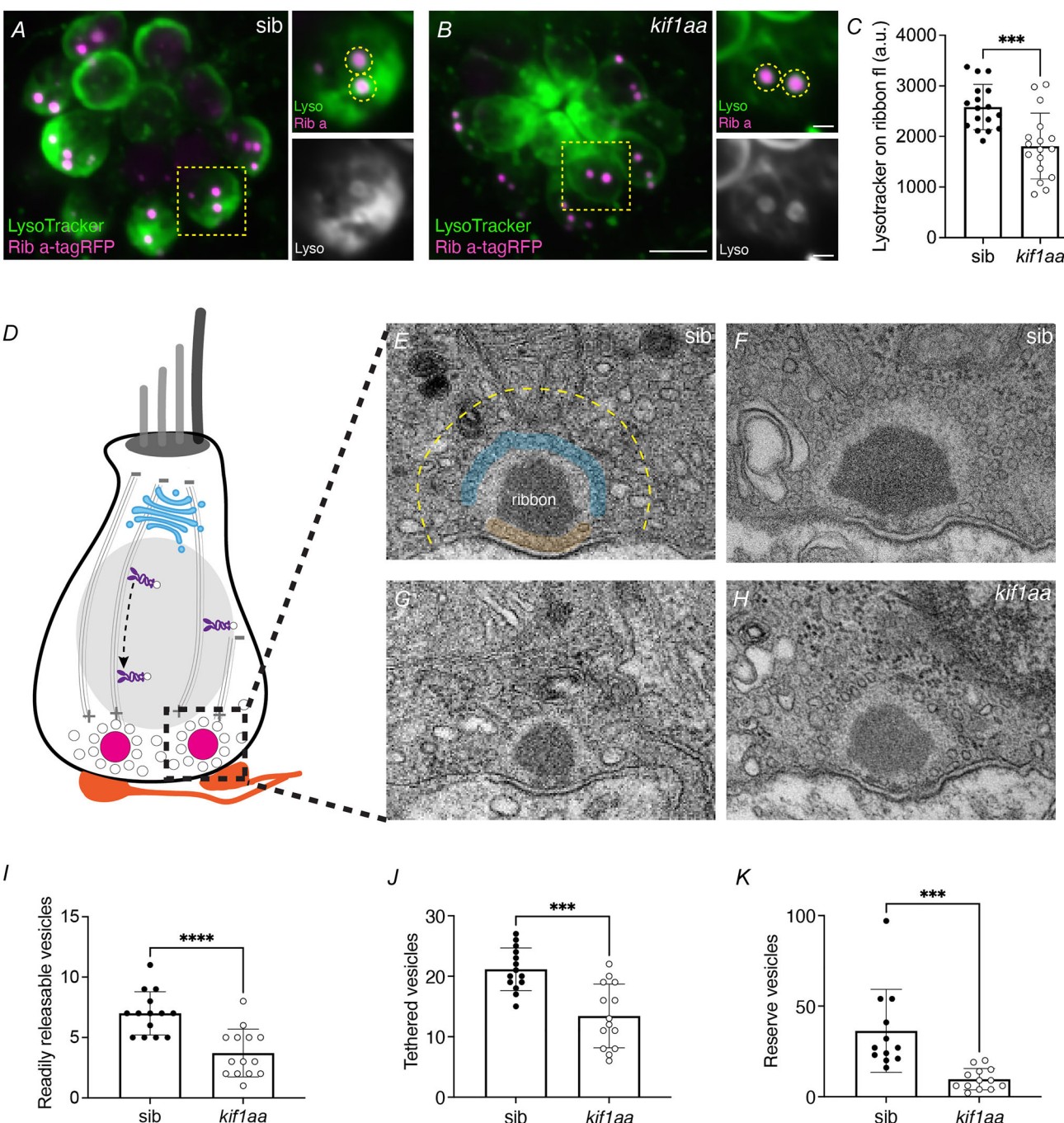

**Figure 7. Kif1aa is important to maintain synaptic vesicles at ribbon synapses**

*A,B*, example live images of neuromasts at 5 dpf in a *kif1aa* mutant (*B*) or sibling control (*A*) at 5 dpf. Synaptic vesicles are labelled with LysoTracker Green (green), and ribbons are labelled with Rib a-TagRFP (magenta). The yellow dashed lines in *A* and *B* were used to create the insets shown to the side. The top inset image shows the base of this single hair cell labelled with Lysotracker (green) and Rib a-TagRFP (magenta), while the bottom inset shows just LysoTracker (grey). Insets are only a partial projection (single cell) of the larger image. The yellow dashed circles indicated ROIs used to quantify LysoTracker intensity at ribbons. *C*, the average LysoTracker intensity at ribbons is significantly lower in *kif1aa* mutants compared with sibling controls (control: 2580 ± 450; *kif1aa*: 1810 ± 650, *n* = 10 control and *kif1aa* neuromasts, unpaired *t* test, *P* = 0.0003). *D*, schematic of an individual hair cell with a ribbon (black dashed box) like ones examined for TEM. *E–H*, representative TEM images of ribbon synapses from *kif1aa* mutants (*G,H*) or wild-type sibling controls (*E,F*) at 5 dpf. In *E* 'tethered vesicles' are in blue, 'readily releasable vesicles' in orange, while 'the reserve pool' within 200 nm of the ribbon is encompassed by the

yellow dashed line. *I-K*, all synaptic-vesicle pools are reduced in *kif1aa* mutants compared with sibling controls (*I*, readily releasable, control: $7 \pm 1.8$; *kif1aa*: $3.7 \pm 2.0$, unpaired *t* test, $P < 0.0001$; *J*, tethered, control: $21.1 \pm 3.5$; *kif1aa*: $13.4 \pm 5.3$, unpaired *t* test, $P = 0.0001$; *K*, reserve pool, control: $36.3 \pm 23.0$; *kif1aa*: $9.7 \pm 5.9$, unpaired *t* test, $P = 0.0005$). $n = 14$ control and *kif1aa* ribbons in *I–J*, and $n = 12$ control and 13 *kif1aa* ribbons in *K*. Scale bars in *A* = 5 μm, inset is 0.1 μm, *F* = 250 nm.

of ribbons. Based on previous studies, we defined these populations as follows: (1) 'vesicles tethered to the ribbon'; (2) 'ready releasable vesicles' located directly beneath the ribbon; and (3) 'reserve pool of vesicles' residing within 200 nm of the ribbon (Fig. 7*E*, zones shaded in blue and orange, and a yellow dashed line) (Lenzi et al., 1999; Nouvian et al., 2006). We found that all three of these synaptic-vesicle populations were reduced in *kif1aa* mutants compared with sibling controls (Fig. 7*I*, readily releasable, control: $7 \pm 1.8$; *kif1aa*: $3.7 \pm 2.0$, unpaired *t* test, $P < 0.0001$; Fig. 7*J*, tethered, control: $21.1 \pm 3.5$; *kif1aa*: $13.4 \pm 5.3$, unpaired *t* test, $P = 0.0001$; Fig. 7*K*, reserve pool, control: $36.3 \pm 23.0$; *kif1aa*: $9.7 \pm 5.9$, unpaired *t* test, $P = 0.0005$). Lastly, we pooled all three of these vesicle populations and found that the total number of ribbon-adjacent vesicles was reduced by 60% in kif1aa mutants (Total ribbon-adjacent vesicles, control: $65.3 \pm 24.4$; *kif1aa*: $26.0 \pm 11.1$, unpaired *t* test, $P < 0.0001$). Altogether, our TEM quantification of synaptic vesicles at ribbons reveals a reduction in all synaptic-vesicle populations in *kif1aa* mutants.

Comparing on-ribbon LysoTracker measurements with TEM-tethered vesicle counts revealed similar reductions in *kif1aa* mutants – 30% and 36%, respectively. These consistent findings suggest that both measurement techniques are reliable for quantifying synaptic vesicles at ribbons. Interestingly, the tethered vesicle counts showed a more modest reduction (36%) compared with the total number of ribbon-adjacent vesicles (60%). This difference suggests that, in *kif1aa* mutants, the synaptic vesicles that reach the cell base closely associate with ribbons rather than being distributed throughout the cell base. Overall, using both methods, we find that synaptic vesicles are significantly depleted at ribbons in *kif1aa* mutants.

### Loss of Kif1aa does not impact mechanosensitive or presynaptic calcium responses

Our results indicate that loss of Kif1aa in lateral-line hair cells dramatically decreases synaptic-vesicle localization at the hair-cell base and at ribbon synapses (Figs 4, 5 and 7). But what impact this impairment in localization has on synaptic-vesicle release at ribbon synapses was unclear. Before examining vesicle release, we performed several control experiments in *kif1aa* mutants and sibling controls.

Prior to assessing vesicle release in lateral-line hair cells, we first examined the localization of voltage-dependent calcium channels ($Ca_V1.3$). $Ca_V1.3$ channel localization, which can impact presynaptic calcium responses and vesicle fusion (Brandt et al., 2005; Trapani & Nicolson, 2011). In addition, calcium channels have been shown to be transported in synaptic precursor vesicles (Petzoldt, 2023). We used immunohistochemistry to examine $Ca_V1.3$ clusters and Rib b to label ribbons (Fig. 8*A–B*). We first quantified the number of paired $Ca_V1.3$-Rib b puncta and found that there were significantly fewer paired puncta per hair cell in *kif1aa* mutants compared with sibling controls (Fig. 8*C*, control: $4.13 \pm 0.39$; *kif1aa*: $3.59 \pm 0.41$; $n = 13$ control and 9 *kif1aa* neuromasts, unpaired *t* test, $P = 0.00550$). Fewer paired $Ca_V1.3$-Rib b puncta per cell is consistent with an overall reduction in the number of complete synapses per cell we observed in *kif1aa* mutants (Fig. 3*D*).

Next, we examined the $Ca_V1.3$ puncta more closely by measuring the 2D area, mean intensity, and mean integrated density of the $Ca_V1.3$ puncta. We did not observe a significant difference in the average area of $Ca_V1.3$ puncta between sibling controls and *kif1aa* mutants (Fig. 8*D*, control: $0.14$ μm$^2$ $\pm 0.02$; *kif1aa*: $0.15$ μm$^2$ $\pm 0.02$; $n = 13$ control and 9 *kif1aa* neuromasts, unpaired *t* test, $P = 0.0953$). We then examined the mean intensity and mean integrated intensity of the $Ca_V1.3$ label, which estimate the density and total number of $Ca_V1.3$ channels present per puncta, respectively. We found that both mean intensity and mean integrated intensity of $Ca_V1.3$ immunolabel per puncta were significantly higher in *kif1aa* mutants compared with controls (Fig. 8*E*, mean intensity, control: $1690 \pm 459$; *kif1aa*: $2960 \pm 1002$, unpaired *t* test, $P = 0.000600$; Fig. 8*F*, integrated intensity, control: $229 \pm 73.6$; *kif1aa*: $444 \pm 180$, unpaired *t* test, $P = 0.000900$; $n = 13$ control and 9 *kif1aa* neuromasts). This indicates that both the density and number of $Ca_V1.3$ channels per presynapse are greater in *kif1aa* mutants compared with sibling controls. Together, our $Ca_V1.3$ immunolabelling experiments indicate that although there are fewer paired $Ca_V1.3$-Rib b puncta in *kif1aa* mutants, on average, a greater number of densely packed $Ca_V1.3$ channels may reside within each $Ca_V1.3$ puncta.

After examining $Ca_V1.3$ channel localization at ribbon synapses, we also performed control experiments to assess hair-cell function. We first examined mechanosensative and presynaptic calcium responses, both of which are activity-dependent events required to drive synaptic-vesicle fusion at ribbon synapses. Assessing mechanosensation was particularly important because

kinesins, like Kif1aa, operate on microtubules, and the kinocilium – a microtubule-based structure – is important for the function of mechanosensory hair bundles in lateral-line hair cells (Kindt et al., 2012). To assess both activities, we performed stimulus-evoked calcium imaging, using a transgenic line that expresses membrane-localized GCaMP6s specifically in hair cells (*myo6b:memGCaMP6s*) (Lukasz & Kindt, 2018; Zhang et al., 2018). This transgenic line can be used to image calcium signals in apical mechanosensory hair bundles, as well as calcium signals at the base of the cell that are associated with presynaptic calcium influx (Fig. 9*A–B* and *D*). To evoke activity, we stimulated lateral-line hair cells with a fluid jet for 500 ms as described previously and read out changes in GCaMPP6s signals (Giese et al., 2023; Kindig et al., 2023). We first examined calcium signals in apical hair bundles to assess mechanosensation. In response to a 500 ms stimulus, we observed similar response magnitudes ($\Delta$F/F GCaMP6s) in *kif1aa* mutants compared with sibling controls (Fig. 9*B,D,F,G*, control: 86.1 $\pm$ 24.2; *kif1aa*: 78.9 $\pm$ 18.7, $n = 8$ control and *kif1aa* neuromasts, unpaired $t$ test, $P = 0.514$). From our calcium imaging experiments, we find that mechanosensation is intact in lateral-line hair cells of *kif1aa* mutants.

After assessing mechanosensation, we used the same transgenic line (*myo6b:memGCaMP6s*) and stimulation approach to examine presynaptic calcium responses at the hair-cell base. In response to a 500 ms stimulus, we observed similar presynaptic-response magnitudes ($\Delta$F/F GCaMP6s) in *kif1aa* mutants compared with sibling controls (Fig. 9*C,E,H,I*, control: 122 $\pm$ 57.6; *kif1aa*: 124 $\pm$ 51.5, $n = 8$ control and *kif1aa* neuromasts, unpaired $t$ test, $P = 0.930$). Overall, our data indicate that both mechanosensitive and presynaptic responses in lateral-line hair cells are largely normal in *kif1aa* mutants.

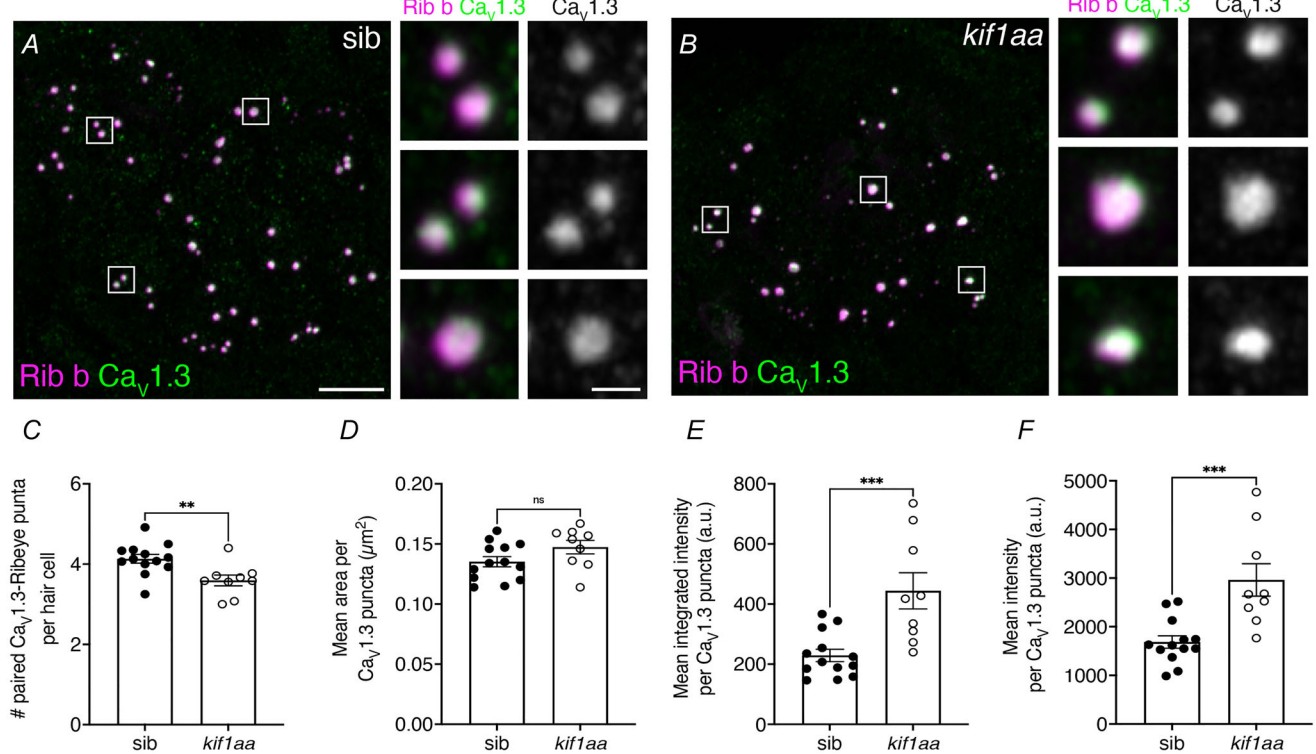

**Figure 8. There are fewer paired Cav1.3-Rib b puncta in *kif1aa* mutants**
*A,B*, example immunostain of neuromasts at 5 dpf in *kif1aa* mutants (*B*) or sibling control (*A*). Ribeye b (Rib b) labels ribbons or presynapses, and Ca$_V$1.3 labels calcium channels. Insets to the side show high magnification images of individual synapses. *C*, the number of paired Rib b-Ca$_V$1.3 puncta is significantly reduced in *kif1aa* mutants compared with sibling controls (control: 4.13 $\pm$ 0.39; *kif1aa*: 3.59 $\pm$ 0.41; $n = 13$ control and 9 *kif1aa* neuromasts, unpaired $t$ test, $P = 0.00550$). *D*, the average size of Ca$_V$1.3 puncta was similar between sibling controls and *kif1aa* mutants (control: 0.14 $\pm$ 0.02; *kif1aa*: 0.15 $\pm$ 0.02, unpaired $t$ test, $P = 0.0953$). *E–F*, the mean intensity and integrated intensity of Ca$_V$1.3 puncta were significantly higher in *kif1aa* mutants compared with controls (*E*, mean intensity, control: 1685 $\pm$ 459; *kif1aa*: 2961 $\pm$ 1002, unpaired $t$ test, $P = 0.000600$; *F*, integrated intensity, control: 230 $\pm$ 73.6; *kif1aa*: 444 $\pm$ 180, unpaired $t$ test, $P = 0.0009$). $n = 13$ control and 9 *kif1aa* neuromasts in *C–F*. Scale bars in *A* = 5 μm, inset is 0.5 μm.

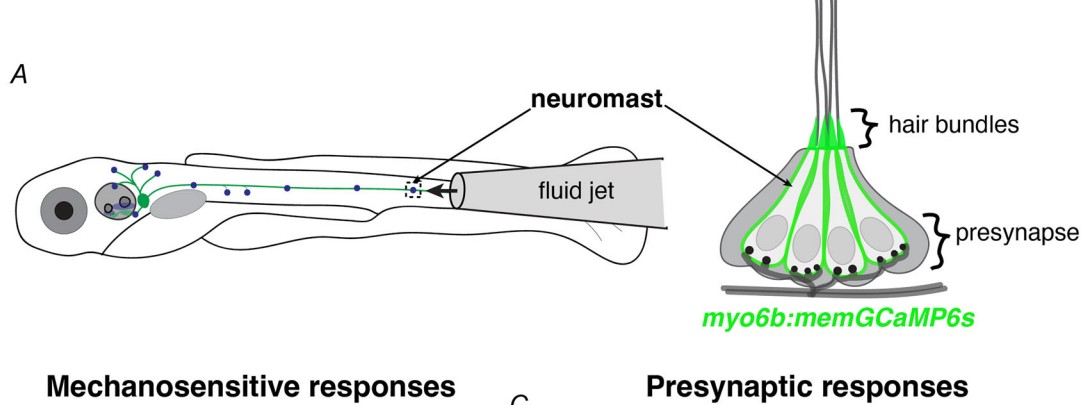

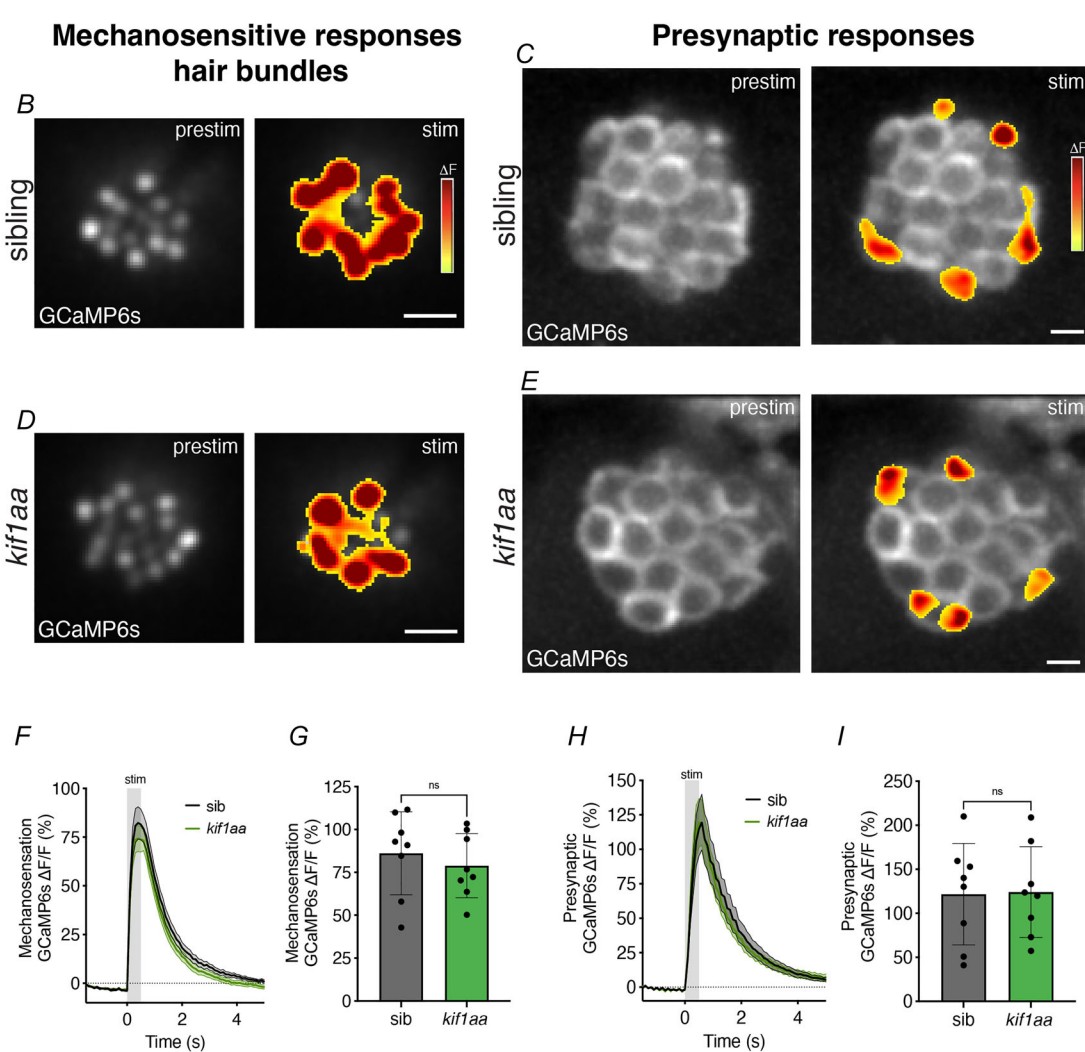

**Figure 9. *Kif1aa* mutants have normal mechanosensitive and presynaptic responses**
*A*, overview of the scheme used to assess evoked calcium responses in lateral-line hair cells. A fluid jet is used to deliver flow stimuli to lateral-line neuromasts. A membrane-localized GCaMP6s (*myo6b:memGCaMP6s*, green) expressed in hair cells is used to measure fluid jet-evoked calcium signals in apical hair bundles or presynaptic calcium signals at the cell base. *B–E*, top-down images show optical planes of memGCaMP6s in neuromast hair bundles (*B,D*) or at the presynapse (*C,E*). Heatmaps show spatial representations of Δ GCaMP signals during evoked mechanosensitive (*B,C*) and presynaptic (*D,E*) activity during a 500 ms stimulation (stim) compared with pre-stimulus (prestim) in sibling controls and *kif1aa* mutants. *F–I*, traces show the average mechanosensitive (*F*) and calcium presynaptic (*H*) calcium responses in sibling control and *kif1aa* mutant hair cells (*n* = 8 neuromasts). Dot plots show that the average mechanosensitive (*G*) and presynaptic (*I*) calcium responses are similar in sibling control and *kif1aa* mutant hair cells (*G*, control: 86.1 ± 24.2, *kif1aa*: 78.9 ± 18.7, *n* = 8 control and *kif1aa* neuromasts,

unpaired $t$ test, $P = 0.514$; *I*, control: $121.7 \pm 57.6$, *kif1aa*: $124.1 \pm 51.5$, $n = 8$ control and *kif1aa* neuromasts, unpaired $t$ test, $P = 0.930$, 5 dpf). Scale bars in *B–E* = 5 μm.

## Loss of Kif1aa alters spontaneous release and postsynaptic calcium responses

Our initial calcium imaging experiments suggest that presynaptic responses in *kif1aa* mutants are relatively normal (Fig. 9*C*). Because *kif1aa* mutants fail to enrich synaptic vesicles at the presynaptic AZ, we hypothesized that vesicle release could still be impaired despite normal presynaptic calcium responses. Therefore, we used additional functional approaches to examine the post-synaptic outcome of fewer synaptic vesicles.

Similar to hair cells in other species, zebrafish lateral-line hair cells release glutamate-filled synaptic vesicles at rest. Established work has shown that this release generates spontaneous spikes in neurons of the posterior lateral-line ganglion (pLLg) neurons that innervate posterior lateral-line neuromasts (Fig. 1*A*; Trapani & Nicolson, 2011). We used *in vivo* loose-patch electrophysiology recordings to measure spontaneous spikes in pLLg neurons (Fig. 10*A*). To quantify the rate of spontaneous spiking, we measured the number of spikes per minute over a 5 min window as previously described (Sheets et al., 2017). We observed a marked 80% decrease in the average afferent spike rate in *kif1aa* mutants compared with sibling controls (Fig. 10*B–C*, control: $208 \pm 148$, *kif1aa*: $42.03 \pm 19.70$; $n = 8$ control and 13 *kif1aa* cells, unpaired $t$ test, $P = 0.0007$). This indicates that a reduction in synaptic vesicles at the presynapse in *kif1aa* mutants results in fewer vesicles released at rest.

To assess evoked postsynaptic activity, we used calcium imaging. For our imaging, we used a transgenic line that expresses GCaMP6s specifically in neurons in the pLLg (*en.sill,hsp70l:GCaMP6s*) (Zhang et al., 2018). We stimulated hair cells with a fluid jet and used this line to measure evoked calcium signals in the afferent terminal beneath lateral-line hair cells as previously described (Fig. 10*D–F*; Kindig et al., 2023). In responses to a 500 ms stimulus, we observed that postsynaptic calcium responses ($\Delta$F/F GCaMP6s) in *kif1aa* mutants were dramatically reduced compared with sibling controls (Fig. 10E–H, control: $82.6 \pm 27.9$; *kif1aa*: $27.2 \pm 24.7$, $n = 10$ control and 9 *kif1aa* neuromasts, unpaired $t$ test, $P = 0.0003$). Together, our postsynaptic electrophysiology and calcium imaging indicate that loss of Kif1aa and fewer synaptic vesicles leads to impaired release of synaptic vesicles.

## Kif1aa is required for normal rheotaxis behaviour but not acoustic startle behaviour

After verifying that loss of Kif1aa impaired synapse function in lateral-line hair cells, we assessed what impact this impairment had on behaviour. We examined two hair cell-mediated behaviours in zebrafish: the acoustic-vibrational startle response and rheotaxis. The acoustic-vibrational startle response is a well-characterized zebrafish behaviour in which a rapid escape reflex is elicited by strong acoustic stimuli. In zebrafish, this reflex can be consistently assessed using a tap stimulus delivered to the dish housing the zebrafish. This tap stimulus is thought to broadly stimulate the zebrafish auditory, vestibular and lateral-line systems, and to some extent somatosensory systems (Burgess & Granato, 2007; Granato et al., 1996; Kimmel et al., 1974). Rheotaxis, whereby fish orient into an oncoming current and hold their position to avoid being swept downstream, is a multisensory behaviour where the lateral-line organ is used to sense changes in water flow (Coombs et al., 2020). Loss of lateral-line function in larval zebrafish leads to an inability to station hold (i.e. hold position at the source of oncoming water flow) during bouts of rheotaxis behaviour (Newton et al., 2023).

To assay the acoustic-vibrational startle response in our *kif1aa* mutants, we used a Zantiks automated behavioural system. We presented larvae in a 12-well plate with an acoustic-vibrational tap stimulus at three levels of intensity (level 3 highest, level 1 lowest), with five trials per intensity level. Intensity levels were chosen based on the proportion of wild-type fish that consistently startled at a given intensity (level 3: 80%; level 2: 50%; level 1: 30%). Each trial was done with a 2 min ISI to avoid habituation (Marsden & Granato, 2015). Using Zantiks software, larvae were recorded and tracked during the experiment. We found that our *kif1aa* mutants were able to respond to a single non-habituating stimulus at the same rate as sibling controls at each stimulus intensity level (Fig. 11*A*, $n = 38$ control and 13 *kif1aa* larvae, two-way ANOVA, level 3: $P = 0.999$, level 2: $P = 0.999$, level 1: $P = 0.994$, no stimulus: $P = 0.586$, 5 dpf). Because we observed no difference in *kif1aa* mutants in response to a single acoustic-vibrational stimulus, we extended the assay to test whether our mutants could maintain responses to repeated or strong habituating stimuli with short ISIs. We reasoned that the *kif1aa* might deplete their limited number of synaptic vesicles when stimulated with short ISIs, leading to faster habituation than controls. For this paradigm, we first administered three non-habituating stimuli with a 2 min ISI, followed by a train of 30 habituating stimuli with a 5 s ISI as previously described (Marsden & Granato, 2015). We also tested recovery from habituation; here, we administered a single stimulus at 20 s, 40 s, 1 min and 2 min after the last habituating stimulus. Using this paradigm, we found that *kif1aa*

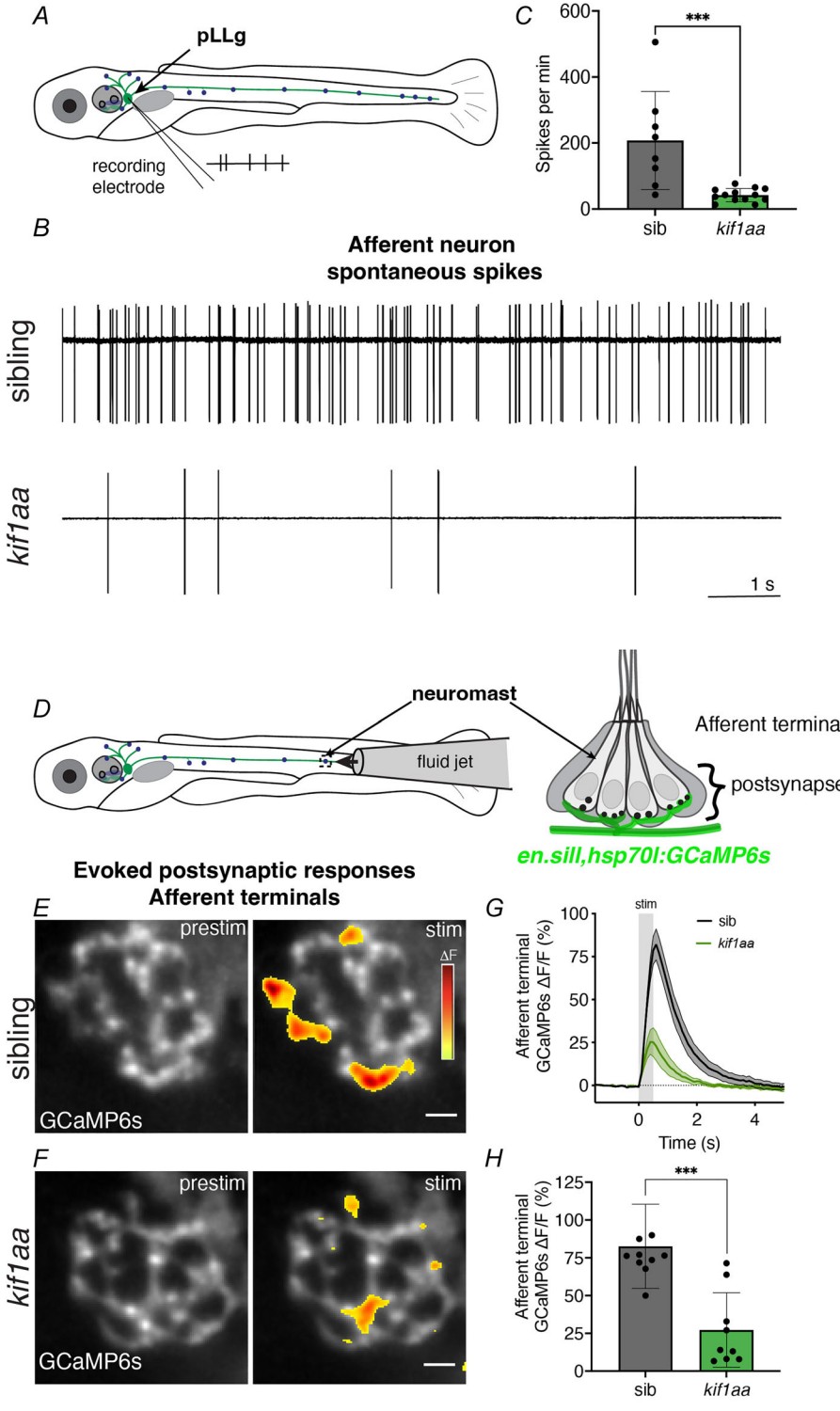

**Figure 10. Afferent neurons in *kif1aa* mutants have fewer spontaneous spikes and reduced evoked responses**

*A*, overview of the scheme used to record spontaneous spiking from afferent cell bodies in the posterior lateral-line ganglion (pLLg). Glass pipettes pulled with long tapers were used to record extracellularly in a loose-patch configuration. *B*, representative 10 s traces show spiking in pLLg neurons in sibling control (top) and *kif1aa* mutant (bottom). *C*, quantification shows that the average number of spikes per minute in *kif1aa* mutants is significantly lower than sibling controls (control: 208 ± 149, *kif1aa*: 42.0 ± 19.7; *n* = 8 control and 13 *kif1aa* cells, unpaired *t* test, *P* = 0.0007, 3–6 dpf). *D*, overview of the scheme used to assess evoked calcium responses in the afferent

terminals beneath lateral-line hair cells. A fluid jet is used to deliver flow stimuli to lateral-line neuromasts. A transgenic line (*en.sill,hsp70l:GCaMP6s*) expressed in posterior lateral-line afferents is used to measure fluid jet-evoked GCaMP6s calcium signals in afferent terminals beneath neuromasts. *E–F*, top-down images show optical planes of GCaMP6s in afferent terminals. Heatmaps show spatial representations of $\Delta$ GCaMP signals in afferent terminals during a 500 ms stimulation (stim) compared with pre-stimulus (prestim) in sibling control and *kif1aa* mutants. *G*, traces show the average response in the afferent terminal in sibling control and *kif1aa* mutants (*n* = 10 control and 9 *kif1aa* neuromasts). *H*, dot plot shows that the average response in the afferent terminal is significantly lower in *kif1aa* mutants compared with sibling control (control: 82.6 ± 27.9, *kif1aa*: 27.2 ± 24.7; *n* = 10 control and 9 *kif1aa* neuromasts, unpaired *t* test, *P* = 0.0003, 4–5 dpf). Scale bars in *E,F* = 5 μm and 1 s in *B*.

mutants habituate at the same rate as sibling controls (Fig. 11*B*, *n* = 38 control and 25 *kif1aa* larvae, two-way ANOVA with multiple comparisons across stimulus and genotype, *P* = 0.545). In addition, the rate of recovery after habituation was not significantly different between *kif1aa* mutants and sibling controls (Fig. 11*B*, *n* = 38 control and 25 *kif1aa* larvae, two-way ANOVA with multiple comparisons across stimulus and genotype, *P* = 0.620). Overall, our acoustic-vibrational startle assays indicate that loss of Kif1aa does not impair this escape reflex.

The acoustic-vibrational startle response is a fast reflexive behaviour that may not require extensive synaptic-vesicle release. Therefore, we examined rheotaxis behaviour, a more sustained behaviour that is reliant on lateral-line hair cells, which had dramatically fewer synaptic vesicles in *kif1aa* mutants. To evaluate rheotaxis behaviour, we used a custom microflume to present laminar flow to zebrafish at 6 dpf, at a rate of ~10 mm s$^{-1}$ (Fig. 11*C*; Baker & Montgomery, 1999; Newton et al., 2023). We recorded the behaviour of larvae with an overhead camera 10 s before flow (pre-stimulus) and for 20 s during flow stimulus using infrared (850 nm) illumination to eliminate visual cues. We measured spatial use of the chamber, distance travelled and rheotaxis events (orienting in flow) during flow. We first examined the spatial use of the flow chamber and found that when performing rheotaxis, wild-type siblings were predominantly positioned in the space at the front of the arena, in the region of the highest flow (Fig. 11*D*; white dashed box). In contrast, *kif1aa* mutants cumulatively show less positioning in high-flow regions, indicating a reduced ability to station hold (Fig. 11*D*). During bouts of rheotaxis, we also observed that *kif1aa* mutants accumulated along the side of the arena in a small lateral gradient in the laminar flow field (Fig. 11*D*; orange dashed box). Together, these observations indicate that *kif1aa* mutants have an impaired ability to maintain position during rheotaxis. To determine why *kif1aa* mutants showed a reduced station holding, we evaluated whether there were differences in the metrics of rheotaxis behaviour for the first (0–10 s) and last half (10–20 s) of stimulus. When we examined distance travelled during rheotaxis, we found that during the last half of the stimulus, *kif1aa* mutants travelled a reduced total distance compared with wild-type siblings (Fig. 11*E*,

0–10 s, control: 43.3, *kif1aa*: 37.3, *P* = 0.532; 11–20 s, control: 50.1, *kif1aa*: 35.1, *P* = 0.0728, *n* = 43 controls and 30 *kif1aa* larvae). Further, we observed a trend toward fewer rheotaxis events in *kif1aa* mutants during the last half of stimulus, though it did not reach significance (Fig. 11*F*, 0–10 s, control: 3.21, *kif1aa*: 3.03, *P* = 0.532; 11–20 s, control: 4.53, *kif1aa*: 3.70, *P* = 0.0728, *n* = 43 controls and 30 *kif1aa* larvae). Importantly, we did not observe a difference in total distance travelled before the onset of flow (Fig. 11*E*). In addition, we found no significant difference in the mean duration of each rheotaxis event in *kif1aa* mutants compared with wild-type siblings (adjusted *P* value = 0.862). Together, these later measurements indicate that loss of Kif1aa impairs rheotaxis behaviour rather than overall motor function.

Taken together, our acoustic-vibrational startle and rheotaxis assays demonstrate that while *kif1aa* mutants can respond normally to discrete acoustic-vibrational stimuli, their ability to effectively station hold during rheotaxis behaviour in sustained flow stimuli is impaired. This suggests that a large population of synaptic vesicles is important for behaviours that require sustained neurotransmission.

## Discussion

Our work with zebrafish demonstrates that both Kif1a and an intact microtubule network are essential to localize synaptic vesicles properly at the presynaptic AZ. Our functional assays reveal that hair cells lacking Kif1aa release fewer synaptic vesicles, leading to impaired afferent responses. Importantly, deficits in hair-cell vesicle release result in impaired station holding during rheotaxis – a behaviour mediated by the lateral-line sensory system. Altogether, our work highlights a new pathway that ensures an adequate supply of synaptic vesicles reaches the cell base to maintain high rates of release at specialized ribbon synapses.

### Accessory machinery and cargo in transport vesicles

Our work demonstrates that lateral-line hair cells lacking Kif1aa have less enrichment of synaptic markers (Figs 4 and 5) and fewer synaptic vesicles at the presynaptic AZ

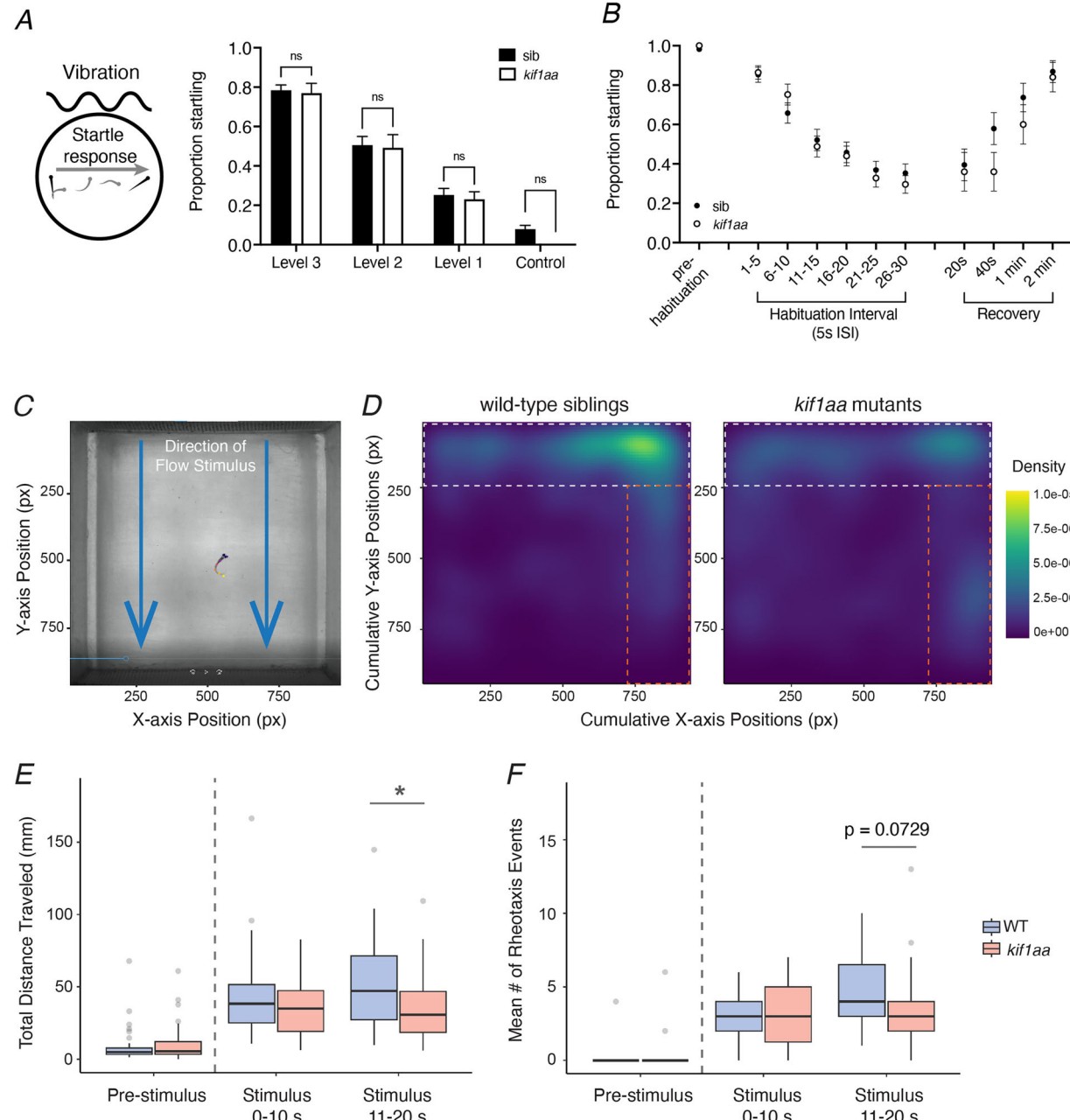

**Figure 11. Station holding within flow, but not acoustic startle is impaired in *kif1aa* mutants**

*A*, a vibrational-acoustic tap stimulus was used to evoke an escape response in sibling control and *kif1aa* mutant larvae at three levels of decreasing intensity. Five stimuli per intensity level were administered, and the proportion of times each animal responded was plotted. No significant difference between *kif1aa* mutants and sibling controls was observed at any intensity tested (control: 0.78 ± 0.16 (Level 3), 0.51 ± 0.27 (Level 2), 0.25 ± 0.21 (Level 1), 0.08 ± 0.12 (no stimulus); *kif1aa*: 0.77 ± 0.18 (Level 3), 0.49 ± 0.24 (Level 2), 0.23 ± 0.14 (Level 1), 0 ± 0 (no stimulus); *n* = 38 control and 13 *kif1aa* larvae, two-way ANOVA with multiple comparisons, *P* = 0.999 (Level 3–2), 0.994 (Level 1) and *P* = 0.586 for no stimulus control; 5 dpf). *B*, an acoustic startle habituation assay was used to assess whether *kif1aa* mutants respond to repeated stimuli. Here a series of 30 acoustic-vibrational tap stimuli were delivered every 5 s. This assay shows that *kif1aa* mutants do not habituate or recover after habituation at a significantly different rate compared with sibling controls (*n* = 38 control and 25 *kif1aa* larvae, two-way ANOVA with multiple comparisons, habituation: genotype × stimulus, *P* = 0.545 (and no significant difference at any interval); recovery: 40 s, *P* = 0.318; 1 min, *P* = 0.721; 2 min, *P* = 0.997). *C*, top-down view of the working section of the microflume apparatus. Blue arrows indicate the direction of water flow. A larval fish performing behaviour is

included for scale. *D*, two-dimensional heat maps showing spatial use/cumulative positioning during flow stimulus. Wild-type siblings (*D*) predominantly maintain position in the space at the front of the arena (white dotted lines) in the strongest part of the flow. Cumulative positioning in the source of the flow is reduced for *kif1aa* mutants (white *vs.* orange dotted lines), indicating impaired ability to station hold. *E*, box and whisker plots of total distance travelled during rheotaxis events. Under flow stimulus, the total distance travelled by *kif1aa* mutant larvae was significantly reduced compared with wild-type siblings during the second half (11–20 s) of flow stimulus (control: 50.1 nm; *kif1aa*: 35.1 nm, adjusted *P* = 0.0170 (11–20 s)). *F*, box and whisker plots of the mean number of rheotaxis events. Under flow stimulus, *kif1aa* mutant larvae trended toward fewer rheotaxis events during the last half (11–20 s) of stimulus, though the difference was not significant (control: 4.5 events; *kif1aa*: 3.7 events, adjusted *P* = 0.0728). *n* = 43 wild-type and 30 *kif1aa* mutant larvae were tested.

(Fig. 7). This finding is consistent with studies in neurons in both mice and *C. elegans* where disruptions in Kif1aa orthologues KIF1A/UNC-104 result in significantly fewer synaptic vesicles at presynapses (Hall & Hedgecock, 1991; Okada et al., 1995). In neurons, synaptic-vesicle precursors are made *de novo* in the ER-Golgi network located in the cell soma. After budding from the trans-Golgi, these precursors are carried to presynaptic terminals along axonal microtubules via kinesin motors and mature into synaptic vesicles at the presynapse (Rizzoli, 2014; Santos et al., 2009). Precursor vesicles can contain many presynaptic components, including: Rab3a, CSP, Vglut, Piccolo, Bassoon, SNARE proteins, neurexins and calcium channels (reviewed in Petzoldt, 2023). In our *kif1aa* mutants, we show that several of these cargos, such as Rab3a, CSP and Vglut3, fail to localize properly at the pre-synaptic AZ in hair cells (Fig. 4). Although we did observe more $Ca_V1.3$ channels per presynapse (Fig. 8*F*), we also observed fewer complete synapses (Fig. 3*D*) and fewer Rib b-$Ca_V1.3$ paired puncta (Fig. 8*C*) in *kif1aa* mutants. Fewer complete pairings could be due to a requirement for Kif1aa to transport material (e.g. Ribeye and $Ca_V1.3$) during synapse formation. This is consistent with our previous work that revealed subtle defects in ribbon formation in lateral-line hair cells lacking Kif1aa (Hussain et al., 2024). Overall, our work on hair cells suggests that Kif1a plays a conserved role in the transport of synaptic material to the presynaptic AZ of neurons and hair cells.

An important part of transport is not only the kinesin motor moving cargo along microtubules but also adaptors that link specific cargos to motors. Work in neurons suggests that the KIF1A motor relies on the adaptor protein MAP kinase activating death domain (MADD) to transport synaptic-vesicle precursors. Specifically, MADD is thought to provide a linkage between RAB3A present on synaptic-vesicle precursors and KIF1 (Hummel & Hoogenraad, 2021; Niwa et al., 2008). In our study and other work on hair cells, Rab3a has been shown to be a marker of synaptic vesicles in hair cells (Einhorn et al., 2012; Uthaiah & Hudspeth, 2010). Along with other synaptic markers, we found that in *kif1aa* mutants, Rab3a immunolabel was not enriched at the hair-cell presynapse (Fig. 4*K–L*). Based on this result and work in neurons, it is possible that Rab3a is part of the link that is required to facilitate the transport of new synaptic-vesicle pre-cursors from the Golgi in hair cells. In the future, it will be interesting to examine the localization of various synaptic-vesicle markers in *rab3a* mutants or to lesion other transport partners, such as Madd, to determine whether they are also required to accumulate synaptic vesicles at the presynapse in hair cells.

## Differences in Vglut3 localization in inner ear hair cells

In *kif1aa* mutants, we observed a clear synaptic-vesicle defect in lateral-line hair cells, characterized by the loss of Vglut3 enrichment at the cell base (Fig. 4*E–H*). However, Vglut3 localization defects in inner ear hair cells were more complex. In the cristae, zebrafish have been shown to have two main hair-cell types: tall, columnar cells and short, flask-like cells (Smith et al., 2020; Zhu et al., 2020). In sibling controls, Vglut3 was enriched at the cell base only in tall cells, and we were unable to detect Vglut3 in short cells (Fig. 5*A*). Similar to lateral-line hair cells, in *kif1aa* mutants, Vglut3 was not enriched at the base of tall cells (Fig. 5*A–C*). This indicates that although *kif1ab* mRNA is expressed in hair cells within the cristae (Fig. 2*D*), Kif1ab is insufficient to compensate for the loss of Kif1aa in the context of synaptic-vesicle localization. Interestingly, tall and short hair cells in zebrafish cristae are proposed to be analogous to mammalian type I and type II vestibular hair cells, respectively (Smith et al., 2020; Zhu et al., 2020). In older mice, there is some evidence that type II, but not type I, vestibular hair cells have a strong Vglut3 immunolabel (Schraven et al., 2012), consistent with our observations in zebrafish. The reduced Vglut3 labelling in type 1 cells may be due to their reliance on non-quantal neurotransmission, which does not require synaptic vesicles and glutamate (Govindaraju et al., 2023).

In contrast to the cristae, we found that all hair cells in the anterior macula of sibling controls displayed Vglut3 immunolabelling. However, similar to the cristae, Vglut3 deficits were observed in the anterior macula of *kif1aa* mutants (Fig. 5*D–F*). While Vglut3 remained enriched at the cell base in both sibling controls and *kif1aa* mutants, the overall levels of Vglut3 immunolabel were reduced in the anterior macula in *kif1aa* mutants (Fig. 5*F*). This suggests that Kif1ab may partially compensate for the loss

of Kif1aa in macular hair cells with regards to Vglut3 enrichment at the cell base. This partial enrichment might be sufficient to maintain the normal acoustic-vibrational responses observed in *kif1aa* mutants (Fig. 11*A–B*). Further studies are needed to explore the roles of Kif1aa and Kif1ab in synaptic-vesicle transport within hair-cell sensory epithelia.

## Mechanisms to maintain synaptic-vesicle populations at the presynapse

We found that fewer synaptic vesicles in *kif1aa* mutants were accompanied by impaired spontaneous and evoked activity in lateral-line afferent neurons (Fig. 10). But despite a dramatic reduction in synaptic vesicles and activity, some synaptic vesicles were still present at ribbon synapses, and activity was not completely abolished (Figs 7, 10). Based on these results, it is likely that there are alternate mechanisms in play that ensure synaptic vesicles reach the presynapse. For example, other kinesin motors could work in tandem with Kif1a to transport synaptic-vesicle precursors. In support of this idea, immunolabelling results have shown that another kinesin, KIF1A, colocalizes with ribbons (Michanski et al., 2019). Alternatively, it is possible that synaptic-vesicle precursors may simply reach the presynapse through diffusion. Studies on ribbon synapses suggest that synaptic vesicles are able to freely diffuse within the cytosol until affixing to the ribbon (Holt et al., 2004; LoGiudice & Matthews, 2009). Based on these diffusion studies, it is possible that synaptic and precursor vesicles in *kif1aa* mutants diffuse from their site of origin near the Golgi until they eventually encounter and bind to a ribbon. In this scenario, although diffusion is less efficient than directed transport, it may allow enrichment of enough synaptic vesicles to explain the residual function at synapses in *kif1aa* mutants.

In many types of neurons, especially those with long axons, it is essential that new synaptic-vesicle precursors are actively transported, as presynaptic terminals are located at considerable distances from the cell soma. In these circumstances, the diffusion of synaptic-vesicle precursors is not practical. In addition to *de novo* synthesis and transport of synaptic material from the Golgi to the presynapse, at both neuronal synapses and ribbon synapses, endocytosis and local synaptic-vesicle recycling also function to maintain synapse function. For example, work in hippocampal neurons has demonstrated that robust recycling of synaptic vesicles can maintain stable and consistent synaptic release even during high frequency and sustained stimulation (Gallimore et al., 2023). In hair cells, multiple forms of endocytosis have been characterized at ribbon synapses, including clathrin-mediated endocytosis and bulk endocytosis;

there is also evidence of kiss-and-run fast endocytosis (Neef et al., 2014). In the future, it will be important to more carefully explore whether endocytic pathways are disrupted in hair cells lacking Kif1aa.

## Impact of Kif1a loss on zebrafish behaviour and in human patients

In humans, mutations in the *KIF1A* gene lead to a rare inherited condition collectively known as KIF1A-associated neurological diseases (KANDs) (Nair et al., 2023). Because *KIF1A* is expressed broadly in the nervous system, there is extensive damage to neurons in the brain and spinal cord in KAND patients. Further, while optic nerve atrophy and vision impairment are also a part of KAND, hearing loss is a less well-studied symptom of this disease (Lee et al., 2015; Montenegro-Garreaud et al., 2020; Pennings et al., 2020). Our work suggests that in addition to neurons, KIF1A may also play an important role in sensory hair cells. Our study highlights this role more clearly because zebrafish have two paralogues of mammalian KIF1A, Kif1aa and Kif1ab (Fig. 1*F*). Based on published scRNAseq data (Sur et al., 2023), in our *kif1aa* zebrafish mutants it is likely that Kif1ab compensates for loss of Kif1aa in most neurons. This allowed us to focus on phenotypes relevant to hair cells in the lateral line, which only express *kif1aa* (Fig. 2*B*). Behaviourally, our *kif1aa* mutants have normal acoustic-vibrational startle responses (Fig. 11*A*). Normal startle behaviour could be due to contributions from hair cells in the zebrafish inner ear, which are probably less affected in *kif1aa* mutants because these cells express both *kif1aa* and *kif1ab* (Figs 2*C–F* and 5). In this scenario, Kif1ab may partially compensate for Kif1aa in hair cells of the zebrafish inner ear. Alternatively, it is possible that fewer synaptic vesicles are needed for this reflexive behaviour. Importantly, our rheotaxis experiments demonstrated that *kif1aa* mutants are unable to maintain position in flow (Fig. 11*D*). Further, the deficits in rheotaxis behaviour occurred during the last 10 s of stimulus (Fig. 11*E,F*), suggesting that the inability to maintain synaptic vesicles at hair-cell presynapses may underlie these deficits. In this scenario, the continuous flow applied during our rheotaxis experiments demands a near constant supply of synaptic vesicles, specifically from lateral-line hair cells. Alternatively, it is possible that the rheotaxis deficits observed in *kif1aa* mutants could be attributed to deficits in neurotransmission at the neuromuscular junction, potentially leading to muscle fatigue. However, our data suggest that it is loss of Kif1aa function in lateral-line hair cells that leads to impaired station holding during prolonged flow (Fig. 11*D,E*). Notably, we found no significant difference in the mean duration of each rheotaxis event between *kif1aa* mutants and wild-type siblings (Stimulus 20 s: genotype, adjusted

*P* value = 0.862). This indicates that the reduction in total distance travelled during rheotaxis is due to fewer events, not muscle fatigue limiting the distance of each event. In future studies, it will be important to rescue Kif1aa in hair cells to ensure that our behavioural phenotypes are due to defects that arise specifically in hair cells.

Overall, our work suggests that inner ear deficits may contribute to phenotypes in human KAND patients. For example, KAND is associated with cerebellar ataxia, and although there is evidence of cerebellar atrophy in KAND patients, it is possible that there are also contributions from the inner ear, particularly the vestibular system, where hair cells must maintain sustained release of synaptic vesicles for proper balance (Paprocka et al., 2023). In the future, it will be important to assess KAND patients carefully for hearing and balance deficits. In addition, our *kif1aa* zebrafish could be used to understand the role KIF1A plays in hair cells or to assess pathogenic variants in KIF1A.

Our current study highlights how an understudied mechanism – Kif1a-based transport – functions to enrich synaptic vesicles at ribbon synapses in hair cells. In the future, it will be important to understand how this transport works alongside recycling and endocytic pathways to supply the near constant demand for synaptic vesicles at ribbon synapses. A powerful way to study synaptic-vesicle dynamics is by using live imaging and sensitive readouts of release, such as afferent recordings. The zebrafish lateral line provides an excellent model for applying these approaches in future studies.

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

## Additional information

### Data availability statement

Raw data and code reported from work completed at the NIH have been deposited and are publicly available at: DOI: https://doi.org/10.5061/dryad.pg4f4qs03. For the rheotaxis assay, all R-analysis scripts, details of apparatus construction, and protocols are provided online in the Open-Source Framework repository, https://osf.io/rvyfz/. All other data reported in this paper will be shared by the lead contact upon request. Any additional information required to reanalyse the data reported in this paper is available from the lead contact upon request.

### Competing interests

The authors declare they have no competing interests.

### Author contributions

K.K. and S.D. did the immunohistochemistry, along with the imaging and analysis of fixed samples. K.P. did the *kif1aa* and *kif1ab* RNA-FISH. Z.L. analysed the RNA-FISH, synapse and Ca$_V$1.3 immunolabels. S.D. did the electrophysiology and K.K. did the calcium imaging. Y.S. prepared and imaged samples for TEM. S.D. did the startle assays. K.N., D.L. and L.S. did the rheotaxis assay and analysis. L.S. made the rheotaxis figure. K.K. and S.D. made figures and wrote the manuscript. S.D., K.P., Z.L., J.S., L.S. and K.K. edited and proofread the manuscript.

### Funding

This work was supported by National Institute on Deafness and other Communication Disorders (NIDCD) Intramural Research Program Grants 1ZIADC000085-01 (K.K.), ZICDC000081 (Y.S.), NIDCD Extramural Research Program Grants R01DC016066 (L.S.) and 'Development of Clinician/Researchers in Academic ENT' training grant, award number T32DC000022 (D.L.). The content is solely the responsibility of the authors and does not necessarily represent the official view of the funding sources.

### Acknowledgements

The authors thank Drs Hiu-tung Wong, Catherine Drerup, Elizabeth Cebul and Doris Wu for their thoughtful comments on the manuscript.

### Keywords

hair cells, kinesins, ribbon synapses, sensory systems, synaptic vesicles, zebrafish

### Supporting information

Additional supporting information can be found online in the Supporting Information section at the end of the HTML view of the article. Supporting information files available:

**Peer Review History**

