## [Peer Review History · The Journal of Physiology]

Kif1a and intact microtubules maintain synaptic-vesicle populations at ribbon synapses in zebrafish hair cells

Sandeep David, Katherine Pinter, Keziah-Khue Nguyen, David S Lee, Zhengchang Lei, Yuliya Sokolova, Lavinia Sheets, and Katie S Kindt

DOI: 10.1113/JP286263

Corresponding author(s): Katie Kindt (katie.kindt@nih.gov)

The following individual(s) involved in review of this submission have agreed to reveal their identity: Allison Coffin (Referee #1)

Review Timeline:

Submission Date:	20-May-2024
Editorial Decision:	09-Jul-2024
Revision Received:	23-Aug-2024
Accepted:	05-Sep-2024

Senior Editor: Katalin Toth

Reviewing Editor: Samuel Young

Transaction Report:

Dear Dr Kindt,

Re: JP-RP-2024-286263 "Kif1a and intact microtubules maintain synaptic-vesicle populations at ribbon synapses in zebrafish hair cells" by Sandeep David, Katherine Pinter, Keziah-Khue Nguyen, David S Lee, Zhengchang Lei, Yuliya Sokolova, Lavinia Sheets, and Katie S Kindt

Thank you for submitting your manuscript to The Journal of Physiology. It has been assessed by a Reviewing Editor and by 2 expert referees and we are pleased to tell you that it is acceptable for publication following satisfactory revision.

REVISION CHECKLIST:

- 'Potential Cover Art' for consideration as the issue's cover image

- Appropriate Supporting Information (Video, audio or data set: see https://jp.msubmit.net/cgi-bin/main.plex?form_type=display_requirements#supp).

We look forward to receiving your revised submission.

Yours sincerely,

Katalin Toth
Senior Editor
The Journal of Physiology

REQUIRED ITEMS

- Author photo and profile. First or joint first authors are asked to provide a short biography (no more than 100 words for one author or 150 words in total for joint first authors) and a portrait photograph. These should be uploaded and clearly labelled together in a Word document with the revised version of the manuscript. See Information for Authors for further details.

- Please upload separate high-quality figure files via the submission form.

- Your paper contains Supporting Information of a type that we no longer publish, including supplementary tables and figures. Any information essential to an understanding of the paper must be included as part of the main manuscript and figures. The only Supporting Information that we publish are video and audio, 3D structures, program codes and large data files. Your revised paper will be returned to you if it does not adhere to our Supporting Information Guidelines.

- Please include an Abstract Figure file, as well as the Figure Legend text within the main article file. The Abstract Figure is a piece of artwork designed to give readers an immediate understanding of the research and should summarise the main conclusions. If possible, the image should be easily 'readable' from left to right or top to bottom. It should show the physiological relevance of the manuscript so readers can assess the importance and content of its findings. Abstract Figures should not merely recapitulate other figures in the manuscript. Please try to keep the diagram as simple as possible and without superfluous information that may distract from the main conclusion(s). Abstract Figures must be provided by authors no later than the revised manuscript stage and should be uploaded as a separate file during online submission labelled as File Type 'Abstract Figure'. Please also ensure that you include the figure legend in the main article file. All Abstract Figures should be created using BioRender. Authors should use The Journal's premium BioRender account to export high-resolution images. Details on how to use and access the premium account are included as part of this email.

EDITOR COMMENTS

Reviewing Editor:

This manuscript is focused on deciphering the roles of kinesin motor protein, Kif1A in hair cell function in zebrafish. Both reviewers found the data highly rigorous, innovative and the findings impactful in our understanding of the cellular and molecular mechanisms of sensory transduction. Both reviewers had comments that that manuscript needs additional quantification of data, in particular VGlut3 staining and a thorough rewrite to make the findings and interpretations for the general reader more accessible. These comments should be addressed by carefully revising and rewriting their manuscript based on the positive criticism provided by the reviewers. Furthermore, supplemental Figures and tables need to be moved and incorporated into the main text of the manuscript. Finally, the method of euthanasia and access to food statement appears to be missing the materials and methods section. Per Journal policy this needs to be added.

REFEREE COMMENTS

Referee #1:

This is a rigorously-conducted study that examines the role of kif1a, a kinesin motor protein, on synaptic morphology and physiology in zebrafish sensory hair cells. The study uses a combination of genetic and pharmacologic manipulation along with several measures of synapse structure and function, and the additional of multiple behavioral assays further adds to this strong body of work. I have a few minor suggestions that I think will further enhance this manuscript.

The biggest suggestion is to add rationale (particularly citations) for experimental details, such as drug/reagent concentrations, hair cell stimulation and imaging paradigms, and in particular, details about the startle assay stimulus (what unit of measure is used to determine stimulus intensity, how is intensity quantified, why these ISIs, etc.?).

Also, the writing is bit choppy in the opening paragraph of the introduction and there are a few grammatical errors and missing commas throughout. None of these are significant issues but one more thorough read by the author string is suggested to further improve flow and catch the remaining errors.

A few specific comments.

1. Line 108, the phrase "clusters of hair cells called neuromasts" is not accurate, since the wording implies that the hair cells are called neuromasts, not that neuromasts contain clusters of hair cells as well as other cell types.
2. The findings of differential vglut3 distribution are interesting, both in the cristae and the inner ear maculae. It would be nice to see a few sentences about these results in the discussion within the context of type 1 vs type 2 hair cells as well as potentially tonotopy in the posterior (sacculus) macula. The type 1 vs. type 2 hair cell comparison was raised in the results but not the discussion.
3. In Figure 6 it's unclear how mean intensity and mean integrated intensity differ (in this case, with regards to CaV1.3 punctae).
4. Based on my understanding of this study, the "acoustic startle response" isn't truly an acoustic response because it doesn't use a sound stimulus. I suggest revising to make it clear that this is a vibrational response caused by a tap, and also to include information about tap frequency and intensity. Further, lines 449-450 state that this response primarily relies on the saccule, which is likely not the case for a vibrational stimulus.
5. For the rheotaxis assay, it's possible that the mutants show reduced rheotaxis because of muscle fatigue (or perhaps an impact of neurotransmission at neuromuscular junctions) rather than a reduction in the vesicular pool at ribbon synapses. It's also possible that the startle response didn't differ between mutants and WT siblings because both kif1aa and kif1ab are expressed in the ear, so kif1ab could compensate in the mutant fish. It would help to see these points briefly addressed in the discussion.
6. For Fig. 1, please indicate in the legend what transgenic line is represented by the YFP; that information was in the methods but I didn't see it elsewhere.

Referee #2:

In this study, the authors aim to investigate the possible role of Kif1a at hair cell ribbon synapses. KIF1A a member of the kinesin 3 family, which has been shown to play a critical role in transporting synaptic vesicles to neuronal synapses via a microtubule network. In hair cells, it is currently unknown how synaptic vesicles reach the release site and whether this involves a similar mechanism as in neurons. The authors have hypothesized that KIF1A is involved in this transport process based on the following evidence: 1) recent work from the Kindt lab has shown that microtubules in hair cells grow from their apical portion toward the active zone via the Golgi area; 2) Kif1a has recently been shown to be expressed in hair cells using scRNAseq data. To address the above hypothesis, the authors have used a combination of approaches ranging from electron microscopy, calcium imaging, pharmacology and in vivo behaviour applied to zebrafish, all of which is routinely performed by this well-established and very productive lab. The study successfully determined that vesicle enrichment at hair cell ribbon synapses, and synaptic function, requires KIF1A and an intact microtubule network. Overall, the work is

novel, very well executed and easily to follow, which is in line with the previous work from this lab. Below are some specific points for author's consideration (in order of appearance):

- I believe that there are no Supplementary Figures in JPhysiol. Therefore, any key Supplementary Figures should be included in the main text.
- In the Results section, it would be useful to provide some tangible evidence for the following statement with references "We performed our analyses at 5 dpf, when the majority of the hair cells are mature,.....". One suggestion would be to just state that at 5dpf "the lateral-line system is functional".
- It is not clear to me why the number of ribbons should be reduced in kif1a mutant zebrafish since the protein should only be involved in trafficking vesicles. Some explanation would be useful to the reader.
- As for the lateral line, it would be useful to have a quantification of VGlut3 in the inner ear.
- It would be useful to specify the number of zebrafish used for each set of experiments.
- I personally found the disparity of results between the different approaches quite confusing. Live imaging and immunostaining data (Figures 3 and 4) seem to suggest a strongly reduced number of vesicles at the active zones (70-80%), which also seems to agree with the in vivo recordings (Figure 8). However, TEM data indicate a more modest reduction in vesicle number (about 50%), which is quite surprising considering that the afferent responses are almost completely abolished (Figure 8). Finally, despite the extremely reduced afferent responses, which is similar to the Cdh23 mutant (Trapani and Nicolson, 2010), the behavioural experiments seem to indicate only a mild dysfunction.
- The rationale for investigating mechano-electrical transduction and the induced calcium signals (Figure 7 and 7-S1) is not clear to me, considering that the predicted and observed defects are at the synaptic level. It would be beneficial to introduce the rationale for this experiment so the reader can appreciate the reasoning behind it.

END OF COMMENTS

Confidential Review

20-May-2024

Please find our responses to all requests and comments below in blue.

REQUIRED ITEMS

- Author photo and profile. First or joint first authors are asked to provide a short biography (no more than 100 words for one author or 150 words in total for joint first authors) and a portrait photograph. These should be uploaded and clearly labelled together in a Word document with the revised version of the manuscript. See Information for Authors for further details.

We have created and uploaded and Author photo and profile document.

- Please upload separate high-quality figure files via the submission form.

We have uploaded separate high-quantiles files for each figure.

- Your paper contains Supporting Information of a type that we no longer publish, including supplementary tables and figures. Any information essential to an understanding of the paper must be included as part of the main manuscript and figures. The only Supporting Information that we publish are video and audio, 3D structures, program codes and large data files. Your revised paper will be returned to you if it does not adhere to our Supporting Information Guidelines.

All of our data has been compiled into a set of main figures with no supplemental materials.

- Please include an Abstract Figure file, as well as the Figure Legend text within the main article file. The Abstract Figure is a piece of artwork designed to give readers an immediate understanding of the research and should summarise the main conclusions. If possible, the image should be easily 'readable' from left to right or top to bottom. It should show the physiological relevance of the manuscript so readers can assess the importance and content of its findings. Abstract Figures should not merely recapitulate other figures in the manuscript. Please try to keep the diagram as simple as possible and without superfluous information that may distract from the main conclusion(s). Abstract Figures must be provided by authors no later than the revised manuscript stage and should be uploaded as a separate file during online submission labelled as File Type 'Abstract Figure'. Please also ensure that you include the figure legend in the main article file. All Abstract Figures should be created using BioRender. Authors should use The Journal's premium BioRender account to export high-resolution images. Details on how to use and access the premium account are included as part of this email.

We have created an Abstract Figure file, as well as an accompanying figure legend for our manuscript. There are included within the main article files.

EDITOR COMMENTS

Reviewing Editor:

This manuscript is focused on deciphering the roles of kinesin motor protein, Kif1A in hair cell function in zebrafish. Both reviewers found the data highly rigorous, innovative and the findings impactful in our understanding of the cellular and molecular mechanisms of sensory transduction. Both reviewers had comments that that manuscript needs additional quantification of data, in particular VGlut3 staining and a thorough rewrite to make the findings and interpretations for the general reader more accessible. These comments should be addressed by carefully revising and rewriting their manuscript based on the positive criticism provided by the reviewers. Furthermore, supplemental Figures and tables need to be moved and incorporated into the main text of the manuscript. Finally, the method of euthanasia and access to food statement appears to be missing the materials and methods section. Per Journal policy this needs to be added.

Based on the journal guidelines and the comments of the reviewers we have revised our manuscript. We have gone through our text to make the findings and interpretations clearer. We also now provide additional quantification of our Vglut3 immunolabel. In addition, all figures have been combined and there are no longer any supplemental figures. Our methods now include a euthanasia and access to food statement. In addition, we have addressed the comments of the reviewers. Our responses are outlined below in blue in a point-by-point manner.

We would like to thank all the reviewers for their time and thoughtful insights to improve our manuscript.

REFEREE COMMENTS

Referee #1:

This is a rigorously-conducted study that examines the role of kif1a, a kinesin motor protein, on synaptic morphology and physiology in zebrafish sensory hair cells. The study uses a combination of genetic and pharmacologic manipulation along with several measures of synapse structure and function, and the additional of multiple behavioral assays further adds to this strong body of work. I have a few minor suggestions that I think will further enhance this manuscript.

The biggest suggestion is to add rationale (particularly citations) for experimental details, such as drug/reagent concentrations, hair cell stimulation and imaging paradigms, and in

particular, details about the startle assay stimulus (what unit of measure is used to determine stimulus intensity, how is intensity quantified, why these ISIs, etc.?).

Clarify drug/reagent concentrations and hair cell stimulation

In our methods and our results sections, we have made it clear how we chose our drug/dye concentrations and our calcium imaging stimulus duration. In both the methods and results references are now also provided.

Results:

“To label synaptic vesicles live, we incubated zebrafish for 15 min with 100 nM of the vital dye LysoTracker. Previous work used this labeling approach in zebrafish to label acidified organelles, including synaptic vesicles, in living lateral-line hair cells (Einhorn *et al.*, 2012).”

“In recent work, we demonstrated that 30-min, 4-hr, and 16-hr treatments with 250-500 nM nocodazole disrupt microtubule networks in lateral-line hair cells (Hussain *et al.*, 2024). For our present study, we incubated larvae in 250 nM nocodazole for 2 hrs to disrupt microtubules, a concentration and incubation duration that did not result in hair-cell death.”

“To evoke activity, we stimulated lateral-line hair cells with a fluid jet for 500-ms as described previously, and read out changes in GCaMP6s signals (Kindig *et al.*, 2023; Giese *et al.*, 2023).”

Methods:

“To destabilize microtubules, *Tg(myo6b:YFP-Hsa.TUBA)^{idc16Tg}* larvae at 5 dpf were incubated in 250 nM nocodazole (Sigma-Aldrich, SML1665) for 2 hours. This concentration of nocodazole has been shown previously to disrupt microtubules in zebrafish hair cells (Hussain *et al.*, 2024).”

“After 2 hours of nocodazole treatment, 100 nM LysoTracker Red DND-99 (ThermoFisher, L7528) was added to the media for 15 min. This concentration and duration has been used previously to label synaptic vesicles in zebrafish hair cells (Einhorn *et al.*, 2012).”

Details about the startle assay stimulus

We have discussed this issue with Zantiks company that manufactures the startle apparatus and have done our best to come up with a way to estimate the intensity of the tap stimulus in the apparatus. Overall, we believe that this additional information will aid with reproducibility. We now state the following in our methods:

“We used 4 different levels of intensity (1-4, increasing in intensity), with level 4 as the highest intensity stimulus. Based on previous literature, vibrations between 100-1000 Hz elicit short latency startle responses in zebrafish larvae (Burgess & Granato, 2007; Beppi *et al.*, 2021). To deliver the acoustic-vibrational stimulus, the solenoid motor in the Zantiks system was set to move by 7.2° (level 4: 4 full steps), 3.6° (level 3: 2 full steps), 1.8° (level 2: 1 full step), and 0.9° (level 1: 1/2 step), with a 4 x 4.25 ms motor speed moving in clockwise

and anticlockwise movements. We used an Optimus+ Red Sound Level Meter (Cirrus Research) to measure the intensity (dB) of each stimulus in the Zantiks chamber. The meter recorded the following sound intensities: 26.4 dB (Level 4), 23.3 dB (Level 3), 17.8 dB (Level 2), and 11.9 dB (Level 1). We chose these stimulus intensities based on the % of wild-type animals responding at each level. Level 4: 100 %; level 3: 80 %; level 2: 50 %; level 1: 30 %. Importantly, we have used this apparatus (at level 3) to parse out zebrafish with complete and moderate defects in the acoustic startle response (Giese *et al.*, 2023).”

Why a 5 s ISI. We tried several different ISIs, including 10 and 20 s ISIs, but these paradigms did not show habituation in wild-type animals. We chose 5 s ISI because wildtype animals showed habituation using our apparatus, and because previous work studying habituation used this ISI (Marsden & Granato, 2015). This previous work is now stated clearly in our results and methods.

Results:

“For this paradigm , we first administered 3 non-habituating stimuli with a 2-minute ISI, followed by a train of 30 habituating stimuli with a 5-second ISI as previously described (Marsden & Granato, 2015).”

Methods:

“For our habituation and recovery assay, a non-habituating stimulus, followed by a habituating stimulus train, and lastly a recovery stimulus train was performed as previously described (Marsden & Granato, 2015). Similar to previous studies, our non-habituating stimulus was presented 3 times (intensity level 4), with 100 s between trials. This was followed by a habituating train of 30 stimuli (same stimulus intensity), presented with 5 s inter-stimulus interval (ISI), an ISI shown to result in habituation (Marsden & Granato, 2015).”

Also, the writing is bit choppy in the opening paragraph of the introduction and there are a few grammatical errors and missing commas throughout. None of these are significant issues but one more thorough read by the author string is suggested to further improve flow and catch the remaining errors.

Thank you for this input. We have combined the first two paragraph and smoothed them out. In addition, we have done a thorough search for comma inconsistencies.

A few specific comments.

1. Line 108, the phrase "clusters of hair cells called neuromasts" is not accurate, since the wording implies that the hair cells are called neuromasts, not that neuromasts contain clusters of hair cells as well as other cell types.

We have changed this statement to: “the lateral line is made up of neuromasts organs that form in lines along the fish.”

2. The findings of differential vglut3 distribution are interesting, both in the cristae and the inner ear maculae. It would be nice to see a few sentences about these results in the discussion within the context of type 1 vs type 2 hair cells as well as potentially tonotopy in the posterior (saccular) macula. The type 1 vs. type 2 hair cell comparison was raised in the results but not the discussion.

Discussion of Vglut3 in type 1 and type 2 vestibular hair cells:

Great suggestions, in our revision, we have now elaborated on the requirement for Vglut3 in type 1 and type 2 mammalian vestibular hair cells, and for Vglut3 in inner ear hair cells in zebrafish in our discussion.

Discussion:

Differences in Vglut3 localization in inner ear hair cells

In *kif1aa* mutants, we observed a clear synaptic vesicle defect in lateral-line hair cells, characterized by the loss of Vglut3 enrichment at the cell base (Figure 4E-H). However, Vglut3 localization defects in inner ear hair cells was more complex. In the cristae, zebrafish have been shown to have two main hair-cell types: tall, columnar cells and short, flask-like cells (Zhu *et al.*, 2020; Smith *et al.*, 2020). In sibling controls, Vglut3 was enriched at the cell base only in tall cells, and we were unable to detect Vglut3 in short cells (Figure 5A). Similar to lateral-line hair cells, in *kif1aa* mutants, Vglut3 was not enriched at the base of tall cells (Figure 5A-C). This indicates that although *kif1ab* mRNA is expressed in hair cells within the cristae (Figure 2D), *Kif1ab* is insufficient to compensate for the loss of *Kif1aa* in the context of synaptic vesicle localization. Interestingly, tall and short hair cells in zebrafish cristae are proposed to be analogous to mammalian type I and type II vestibular hair cells, respectively (Zhu *et al.*, 2020; Smith *et al.*, 2020). In older mice, there is some evidence that type II, but not type I, vestibular hair cells have strong Vglut3 immunolabel (Schraven *et al.*, 2012), consistent with our observations in zebrafish. The reduced Vglut3 labeling in type 1 cells may be due to their reliance on nonquantal neurotransmission, which does not require synaptic vesicles and glutamate (Govindaraju *et al.*, 2023).

In contrast to the cristae, we found that all hair cells in the anterior macula of sibling controls displayed Vglut3 immunolabeling. However, similar to the cristae, Vglut3 deficits were observed in the anterior macula of *kif1aa* mutants (Figure 5D-F). While Vglut3 remained enriched at the cell base in both sibling controls and *kif1aa* mutants, the overall levels of Vglut3 immunolabel were reduced in the anterior macula in *kif1aa* mutants (Figure 5F). This suggests that *Kif1ab* may partially compensate for the loss of *Kif1aa* in macular hair cells with regards to Vglut3 enrichment at the cell base. This partial enrichment might be sufficient to maintain the normal acoustic vibrational responses observed in *kif1aa* mutants (Figure 11A-B). Further studies are needed to explore the roles of *Kif1aa* and *Kif1ab* in synaptic vesicle transport within hair-cell sensory epithelia.”

Quantification of Vglut3 in inner ear hair cells

We were able to quantify the Vglut3 intensity within tall cells in the zebrafish cristae and the overall expression in the anterior macula (See below-per R2's request). We are less

confident about quantifying our Vglut3 posterior macula immunolabel as it is challenging to image. Therefore, we have removed the posterior macula immunolabel from our manuscript and do not discuss a potential impact on tonotopy in the posterior macula. But in the future, this is something that would be interesting to follow up on!

3. In Figure 6 it's unclear how mean intensity and mean integrated intensity differ (in this case, with regards to Ca_v1.3 punctae).

Great suggestion. We have added the following clarifications.

Results:

“We then examined the mean intensity and mean integrated intensity of the Ca_v1.3 label, which estimate the density and total number of Ca_v1.3 channels present per puncta, respectively. We found that both mean intensity and mean integrated intensity of Ca_v1.3 immunolabel per puncta were significantly higher in *kif1aa* mutants compared to controls This indicates that both the density and number of Ca_v1.3 channels per presynapse are greater in *kif1aa* mutants compared to sibling controls. Together, our Ca_v1.3 immunolabeling experiments indicate that although there are fewer paired Ca_v1.3-Rib b puncta in *kif1aa* mutants, on average, a greater number of densely packed Ca_v1.3 channels may reside within each Ca_v1.3 puncta.”

4. Based on my understanding of this study, the "acoustic startle response" isn't truly an acoustic response because it doesn't use a sound stimulus. I suggest revising to make it clear that this is a vibrational response caused by a tap, and also to include information about tap frequency and intensity. Further, lines 449-450 state that this response primarily relies on the saccule, which is likely not the case for a vibrational stimulus.

In our results section we have clarified the acoustic-vibrational startle assay we used and the sensory systems it is thought to activate.

Results:

“The acoustic-vibrational startle response is a well-characterized zebrafish behavior in which a rapid escape reflex elicited by strong acoustic stimuli. In zebrafish, this reflex can be consistently assayed using a tap stimulus delivered to the dish housing the zebrafish. This tap stimulus is thought to broadly stimulate the zebrafish auditory, vestibular, and lateral-line systems, and to some extent somatosensory systems (Kimmel *et al.*, 1974; Granato *et al.*, 1996; Burgess & Granato, 2007).”

Please see our response above regarding the tap stimulus assay intensity.

5. For the rheotaxis assay, it's possible that the mutants show reduced rheotaxis because of muscle fatigue (or perhaps an impact of neurotransmission at neuromuscular junctions) rather than a reduction in the vesicular pool at ribbon synapses. It's also possible that the startle response didn't differ between mutants and WT siblings because both *kif1aa* and *kif1ab* are expressed in the ear, so *kif1ab* could compensate in the mutant fish. It would help to see these points briefly addressed in the discussion.

Rheotaxis and the potential for muscle fatigue

It is possible that the *kif1aa* mutant rheotaxis defects could be due to muscle fatigue. We have added the following statement to the results and discussion sections:

Results:

“Importantly, we did not observe a difference in total distance traveled before the onset of flow (Figure 11E). In addition, we found no significant difference in the mean duration of each rheotaxis event in *kif1aa* mutants compared to wild-type siblings (adjusted p value = 0.862). Together, these later measurements indicate that loss of Kif1aa impairs rheotaxis behavior rather than overall motor function.”

Discussion:

“Alternatively, it is possible that the rheotaxis deficits observed in *kif1aa* mutants could be attributed to deficits in neurotransmission at the neuromuscular junction, potentially leading to muscle fatigue. However, our data suggest that it is loss of Kif1aa function in lateral-line hair cells that leads to impaired station holding during prolonged flow (Figure 11D,E). Notably, we found no significant difference in the mean duration of each rheotaxis event between *kif1aa* mutants and wild-type siblings (Stimulus 20 s: genotype, adjusted p value = 0.862). This indicates that the reduction in total distance traveled during rheotaxis is due to fewer events, not muscle fatigue limiting the distance of each event. In future studies, it will be important to rescue Kif1a in hair cells to ensure that our behavioral phenotypes are due to defects that arise specifically in hair cells.”

Compensation for Kif1aa by Kif1ab in the inner ear

We have expanded on this in our discussion as follows:

“Behaviorally, our *kif1aa* mutants have normal acoustic-vibrational startle responses (Figure 11A). Normal startle behavior could be due to contributions from hair cells in the zebrafish inner ear which are likely less affected in *kif1aa* mutants because these cells express both *kif1aa* and *kif1ab* (Figure 5, Figure 2C-F). In this scenario Kif1ab may partially compensate for Kif1ab in hair cells of the zebrafish inner ear.”

6. For Fig. 1, please indicate in the legend what transgenic line is represented by the YFP; that information was in the methods but I didn't see it elsewhere.

For this Figure-which is now Figure 2, the legend now states, “The gray label is YFP that is expressed specifically in hair cells by the transgenic line *Tg(myo6b:Cr.ChR2-EYFP)*.”

Referee #2:

In this study, the authors aim to investigate the possible role of Kif1a at hair cell ribbon synapses. KIF1A a member of the kinesin 3 family, which has been shown to play a critical role in transporting synaptic vesicles to neuronal synapses via a microtubule network. In

hair cells, it is currently unknown how synaptic vesicles reach the release site and whether this involves a similar mechanism as in neurons. The authors have hypothesized that KIF1A is involved in this transport process based on the following evidence: 1) recent work from the Kindt lab has shown that microtubules in hair cells grow from their apical portion toward the active zone via the Golgi area; 2) Kif1a has recently been shown to be expressed in hair cells using scRNAseq data. To address the above hypothesis, the authors have used a combination of approaches ranging from electron microscopy, calcium imaging, pharmacology and in vivo behaviour applied to zebrafish, all of which is routinely performed by this well-established and very productive lab. The study successfully determined that vesicle enrichment at hair cell ribbon synapses, and synaptic function, requires KIF1A and an intact microtubule network. Overall, the work is novel, very well executed and easy to follow, which is in line with the previous work from this lab. Below are some specific points for author's consideration (in order of appearance):

- I believe that there are no Supplementary Figures in JPhysiol. Therefore, any key Supplementary Figures should be included in the main text.

Yes, no Supplementary Figures allowed. The Figures have been condensed to include all the key data in 11 main Figures.

- In the Results section, it would be useful to provide some tangible evidence for the following statement with references "We performed our analyses at 5 dpf, when the majority of the hair cells are mature,.....". One suggestion would be to just state that at 5dpf "the lateral-line system is functional".

We now state in the results:

"We performed our analyses at 5 dpf, when the lateral-line system is functional (Suli *et al.*, 2012)."

- It is not clear to me why the number of ribbons should be reduced in kif1a mutant zebrafish since the protein should only be involved in trafficking vesicles. Some explanation would be useful to the reader.

We have added the following statement to our results and discussion for clarity:

Results:

Fewer synapses is consistent our work that demonstrated Kif1aa also plays a subtle role in presynapse assembly (Hussain *et al.*, 2024)."

Discussion:

Fewer complete pairings could be due to a requirement for Kif1aa to transport material (ex: Ribeye and Ca_v1.3) during synapse formation. This is consistent with our previous work that revealed subtle defects in ribbon formation in lateral-line hair cells lacking Kif1aa (Hussain *et al.*, 2024).

- As for the lateral line, it would be useful to have a quantification of VGlut3 in the inner ear.

We have now included quantification of the Vglut3 immunolabel to the best of our abilities.

1. We quantified the enrichment of Vglut3 in the tall cells of the cristae in sibling controls and *kif1aa* mutants. We now show that this enrichment is diminished in *kif1aa* mutants. See Figure 5A-C.
2. We were unable to quantify Vglut3 enrichment at the hair cell base in the macular organs. Instead, we quantified the mean levels of Vglut3 in the anterior macula and show that the overall levels are diminished in *kif1aa* mutants compared to sibling controls. See Figure 5D-F.

- It would be useful to specify the number of zebrafish used for each set of experiments.

In our revision we have created Table 3 that includes the animal counts for each Figure panel. This table is referenced in our Statistics section. In the legends we have kept the N that pertains to the dots in the plots.

- I personally found the disparity of results between the different approaches quite confusing. Live imaging and immunostaining data (Figures 3 and 4) seem to suggest a strongly reduced number of vesicles at the active zones (70-80%), which also seems to agree with the in vivo recordings (Figure 8). However, TEM data indicate a more modest reduction in vesicle number (about 50%), which is quite surprising considering that the afferent responses are almost completely abolished (Figure 8). Finally, despite the extremely reduced afferent responses, which is similar to the *Cdh23* mutant (Trapani and Nicolson, 2010), the behavioural experiments seem to indicate only a mild dysfunction.

Clarifying synaptic vesicle measurements:

The different approaches and domains used to compare synaptic vesicles can get confusing. We now provide additional information to clarify how and why there are different results. For example, measurement at the entire cell base vs. those specifically at the ribbons. We have added the following information to the results:

Results:

“Our live and fixed confocal imaging strongly suggests that synaptic vesicles fail to enrich at the cell base in *kif1aa* mutants (Figure 4, 5). Compared to sibling controls, the overall apex-to-base enrichment of Vglut3 and LysoTracker labels were reduced by 80 and 92 % respectively, in *kif1aa* mutants. To understand how synaptic vesicles are impacted more focally at sites of release, we examined synaptic vesicles at individual ribbons in lateral-line hair cells.”

At the end of this section, we also state:

“Comparing on-ribbon LysoTracker measurements with TEM-tethered vesicle counts revealed similar reductions in *kif1aa* mutants—30 % and 36 %, respectively. These

consistent findings suggest that both measurement techniques are reliable for quantifying synaptic vesicles at ribbons. Interestingly, the tethered vesicle counts showed a more modest reduction (36 %) compared to the total number of ribbon-adjacent vesicles (60 %). This difference suggests that, in *kif1aa* mutants, the synaptic vesicles that reach the cell base closely associate with ribbons rather than being distributed throughout the cell base. Overall, using both methods, we find that synaptic vesicles are significantly depleted at ribbons in *kif1aa* mutants.”

Linking synaptic vesicle measurements to spike numbers:

It is not fully clear to us whether it is the total number of vesicles at the cell base or ribbon adjacent synaptic vesicles that contribute to the spontaneous spike rate. Therefore, we have not discussed this relationship or linked a particular vesicle population to the spontaneous spike rate.

Comparison of spontaneous spikes in *kif1aa* and *cdh23* mutants

It is true that *cdh23* mutants and *kif1aa* mutants have a similar reduction in spontaneous spiking in lateral-line afferents. But in addition to a reduction in spontaneous spikes, there is an absence of evoked afferent activity in *cdh23* mutants. In Trapani et al, *cdh23* mutants show no evoked spikes. In contrast, using calcium imaging we detected normal evoked MET and presynaptic responses (which would not be present in *cdh23* mutants) as well as some evoked activity in the afferent process. It appears that it is these residual evoked responses are sufficient for residual lateral-line function in *kif1aa* mutants.

- The rationale for investigating mechanoelectrical transduction and the induced calcium signals (Figure 7 and 7-S1) is not clear to me, considering that the predicted and observed defects are at the synaptic level. It would be beneficial to introduce the rationale for this experiment so the reader can appreciate the reasoning behind it.

Great idea. We have added this clarifying information.

Results:

“Assessing mechanosensation was particularly important because Kinesins, like Kif1aa, operate on microtubules, and the kinocilium—a microtubule-based structure—is important for the function of mechanosensory hair bundles in lateral-line hair cells”

END OF COMMENTS

The Physiological Society is a company limited by guarantee. Registered in England and Wales, No. 00323575. Registered Office: Hodgkin Huxley House, 30 Farringdon Lane, London, EC1R 3AW, UK. Registered Charity No. 211585.

The Physiological Society and The Journal of Physiology are registered trademarks.

Dear Dr Kindt,

Re: JP-RP-2024-286263R1 "Kif1a and intact microtubules maintain synaptic-vesicle populations at ribbon synapses in zebrafish hair cells" by Sandeep David, Katherine Pinter, Keziah-Khue Nguyen, David S Lee, Zhengchang Lei, Yuliya Sokolova, Lavinia Sheets, and Katie S Kindt

We are pleased to tell you that your paper has been accepted for publication in The Journal of Physiology.

Authors should note that it is too late at this point to offer corrections prior to proofing. Major corrections at proof stage, such as changes to figures, will be referred to the Editors for approval before they can be incorporated. Only minor changes, such as to style and consistency, should be made at proof stage. Changes that need to be made after proof stage will usually require a formal correction notice.

If you would like to receive our 'Research Roundup', a monthly newsletter highlighting the cutting-edge research published in The Physiological Society's family of journals (The Journal of Physiology, Experimental Physiology and Physiological Reports), please click this link, fill in your name and email address and select 'Research Roundup': <https://www.physoc.org/journals-and-media/membernews/>.

Yours sincerely,

Katalin Toth
Senior Editor
The Journal of Physiology

P.S. - You can help your research get the attention it deserves! Check out Wiley's free Promotion Guide for best-practice recommendations for promoting your work at www.wileyauthors.com/eeo/guide. You can learn more about Wiley Editing Services which offers professional video, design, and writing services to create shareable video abstracts, infographics, conference posters, lay summaries, and research news stories for your research at www.wileyauthors.com/eeo/promotion.

IMPORTANT NOTICE ABOUT OPEN ACCESS: To assist authors whose funding agencies mandate public access to published research findings sooner than 12 months after publication, The Journal of Physiology allows authors to pay an Open Access (OA) fee to have their papers made freely available immediately on publication.

You can check if your funder or institution has a Wiley Open Access Account here: <https://authorservices.wiley.com/author-resources/Journal-Authors/licensing-and-open-access/open-access/author-compliance-tool.html>.

EDITOR COMMENTS

Reviewing Editor:

The authors have done an excellent job of responding to the previous critiques. There are no further concerns.

REFEREE COMMENTS

Referee #1:

The authors have done an excellent job addressing reviewer comments.

Referee #2:

The authors have fully addressed my comments. Thank you.

1st Confidential Review

23-Aug-2024